# H4K16ac activates the transcription of transposable elements and contributes to their *cis*-regulatory function

Debosree Pal[1,7], Manthan Patel[1,7], Fanny Boulet[1], Jayakumar Sundarraj[1,2], Olivia A. Grant[1,3], Miguel R. Branco [1], Srinjan Basu [4], Silvia D. M. Santos [5], Nicolae Radu Zabet[1], Paola Scaffidi [5,6] & Madapura M. Pradeepa [1]✉

Mammalian genomes harbor abundant transposable elements (TEs) and their remnants, with numerous epigenetic repression mechanisms enacted to silence TE transcription. However, TEs are upregulated during early development, neuronal lineage, and cancers, although the epigenetic factors contributing to the transcription of TEs have yet to be fully elucidated. Here, we demonstrate that the male-specific lethal (MSL)-complex-mediated histone H4 acetylation at lysine 16 (H4K16ac) is enriched at TEs in human embryonic stem cells (hESCs) and cancer cells. This in turn activates transcription of subsets of full-length long interspersed nuclear elements (LINE1s, L1s) and endogenous retrovirus (ERV) long terminal repeats (LTRs). Furthermore, we show that the H4K16ac-marked L1 and LTR subfamilies display enhancer-like functions and are enriched in genomic locations with chromatin features associated with active enhancers. Importantly, such regions often reside at boundaries of topologically associated domains and loop with genes. CRISPR-based epigenetic perturbation and genetic deletion of L1s reveal that H4K16ac-marked L1s and LTRs regulate the expression of genes in *cis*. Overall, TEs enriched with H4K16ac contribute to the *cis*-regulatory landscape at specific genomic locations by maintaining an active chromatin landscape at TEs.

Dysregulation of TEs and their insertions into gene exons are usually disruptive and have been implicated in cancer and neurological disorders[1,2]. When inserted into noncoding DNA, including introns, they can affect the host gene expression in *cis* or *trans*. Most TEs cannot transpose owing to acquired mutations and epigenetic and post-transcriptional silencing mechanisms (reviewed in refs. 3,4). Transcription of TEs is repressed by DNA methylation, trimethylated histone H3 K9 (H3K9me3), TRIM28 and Krüppel-associated box-containing zinc finger proteins (KRAB-ZFPs), and the human silencing hub (HUSH) complex[5–9]. Apart from these repressive mechanisms, several pluripotency-associated transcription factors (TFs), namely SP1, SP3, LBP9, DUX4, DUX, GATA2 and YY1, are enriched at ERV LTRs, and SOX11, RUNX3 and YY1 are enriched at the 5′ untranslated regions (UTRs; containing promoters) of L1 (reviewed in ref. 10). Interestingly,

[1]Blizard Institute, Faculty of Medicine and Dentistry, Queen Mary University of London, London, UK. [2]Bhabha Atomic Research Centre, Mumbai, India. [3]School of Life Sciences, University of Essex, Colchester, UK. [4]Wellcome-MRC Cambridge Stem Cell Institute, University of Cambridge, Cambridge, UK. [5]Francis Crick Institute, London, UK. [6]Department of Experimental Oncology, European Institute of Oncology, Milan, Italy. [7]These authors contributed equally: Debosree Pal, Manthan Patel. ✉e-mail: p.m.madapura@qmul.ac.uk

most species-specific DNase hypersensitive sites (which are on accessible chromatin) are occupied by remnants of TEs[11,12], suggesting that TEs have been co-opted, becoming tissue- and species-specific *cis*-regulatory elements (CREs). TEs are transiently upregulated during early development[13], in the neuronal lineage[14], and in cancer[1]. The ERV superfamily of LTRs (LTR/ERV) and Alu family of short interspersed nuclear elements (SINE/Alu) often exhibit chromatin features associated with active CREs[15–17] and function either as enhancers to regulate genes in *cis* or act as alternative promoters[15]. The 5′ UTR of L1 repeats are also bound by tissue-specific TFs, are enriched with chromatin features that are associated with CREs[18], and can function as nuclear noncoding RNAs[13,19]; still, it is unclear whether they can act as CREs. Although TEs have been suggested to contribute to nearly one-quarter of the regulatory epigenome[10,13,20,21], the chromatin-based mechanisms contributing to regulatory activity in the vast number of TEs are unclear.

Chromatin features, such as a combination of H3K4me1 and H3K27ac, bidirectional transcription of enhancer RNAs (eRNAs), and accessible chromatin (determined, for example, using the assay for transposase-accessible chromatin with sequencing (ATAC-seq)) are widely used to predict enhancer activity, including for TE-derived enhancers[17,22–25]. Yet the level of H3K27ac does not correlate with and is dispensable for enhancer activity, suggesting that other uncharacterized chromatin features could contribute to regulatory activity[26–28]. H4K16ac and H3K122ac are particularly interesting among many histone acetylations, because they alter chromatin structure directly and increase transcription in vitro[29,30]. H4K16ac and H3K122ac are enriched at enhancers, and they identify new repertoires of active enhancers that lack detectable H3K27ac[27,31]. However, it is challenging to decipher the causal role of specific histone acetylations, as many acetylations, including H3K27ac, are catalyzed by multiple lysine acetyltransferases (KATs), and KATs also have a broad substrate specificity. H4K16ac is an exception, as it is catalyzed explicitly by KAT8 when associated with the MSL complex.

Nevertheless, when KAT8 is associated with non-specific lethal (NSL), it catalyzes H4K5ac, H4K8ac and H4K12ac (refs. [32–35]). In mouse embryonic stem cells (mESCs), KAT8 and H4K16ac mark active enhancers and promoters of genes that maintain the identity of the mESCs[27,36]. Loss-of-function mutations in *KAT8* or *MSL3* lead to reduced H4K16ac levels and are known to cause neurodevelopmental disorders[37,38]. However, the mechanism through which KAT8 containing MSL complex-mediated acetylation of H4K16 contributes to genome regulation during normal development is less clear, especially in the human genome.

Here, we show that H4K16ac is enriched at L1s and LTRs and is depleted at gene promoters, and that H4K16ac regulates transcription across the L1 and ERV LTR superfamily of TEs. TEs marked with acetylations loop with the neighboring genes and regulate their expression. CRISPR interference and genetic deletion of H4K16ac-marked (H4K16ac+) TEs leads to the downregulation of genes in *cis*, demonstrating that H4K16ac+ TEs function as enhancers. Furthermore, depletion of H4K16ac is sufficient for downregulation of L1 and LTRs and genes

linked to these TEs, confirming the significance of H4K16ac-mediated activation of TEs in rewiring the regulatory landscape of a substantial fraction of the mammalian genome.

## Results

We aimed to investigate the role of MSL-mediated H4K16ac in human genome regulation. We performed two to three replicates of cleavage under targets and tagmentation (CUT&Tag)[39] in human embryonic stem cells (H9-hESCs) for histone modifications associated with active regulatory elements (H3K27ac, H3K122ac, H4K12ac, H4K16ac, monomethylated H3 K4 (H3K4me1) and H3K4me3), polycomb repressed domains (H3K27me3) and heterochromatin (H3K9me3) (Extended Data Fig. 1a and Supplementary Table 1). We evaluated overall data quality and similarity among our CUT&Tag replicates (Extended Data Fig. 1a). We generated peaks by merging the replicates, and we used reproducible peaks in at least two replicates to validate our findings (Extended Data Figs. 1 and 2 and Supplementary Tables 2 and 3). To prevent the same reads from mapping to multiple regions in the repeat elements, uniquely mapped CUT&Tag sequencing reads were used for the analyses. Except for the analysis in Figure 5e, we used multi-mapping reads for L1 subfamily-level enrichment analysis.

### H4K16ac and H3K122ac are enriched at TEs

Chromatin-state discovery and genome annotation analysis (Chrom-HMM)[40] of CUT&Tag peaks revealed the expected enrichment of H3K4me1, H3K4me3, H3K27ac and H4K12ac at chromatin features associated with active transcription, including active promoters and enhancers. Intriguingly, H4K16ac and H3K122ac, but not H3K27ac or H4K12ac, were enriched at heterochromatin, insulator and transcription elongation states (Fig. 1a). Further analysis revealed specific enrichment of H4K16ac and H3K122ac at the 5′ UTR of full-length L1s and ERV/LTR elements, compared with gene promoters (Fig. 1b–e). H4K16ac was also detected at gene bodies, consistent with previous findings showing its role in transcription elongation[41] (Fig. 1a,b). Interestingly, however, H4K16ac shows a very low level of enrichment at the gene promoters (Fig. 1b,d,e and Supplementary Table 3), similar to recent chromatin immunoprecipitation and sequencing (ChIP–seq) data in human cell lines[34]. Fifty-one percent of full-length L1s (n = 10,000) have reproducible H4K16ac peaks (Supplementary Table 3), and although H4K16ac and H3K9me3 are enriched at L1s (Fig. 1b), they are anti-correlated at L1 subfamilies (Extended Data Fig. 1c). Less than 10% of the H4K16ac peaks at TEs overlapped with H3K9me3 (Extended Data Fig. 1b). The H3K9me3 level was also lower at H4K16ac peaks that overlapped with L1 5′ UTRs, and the H4K16ac level was lower at L1s with H3K9me3 peaks (Extended Data Fig. 1c). Reanalysis of public ChIP–seq datasets showed enrichment of H4K16ac at the 5′ UTRs of L1s in human brain tissue (Extended Data Fig. 3a)[42]. H4K16ac is also enriched at the 5′ UTRs of L1s in neuroblastoma (SH-SY5Y), erythroleukemia (K562), and transformed dermal fibroblast (TDF) cell lines and mESCs (Extended Data Fig. 3b–e). This analysis suggests that H4K16ac enrichment at TEs is not unique to hESCs, but is conserved in cancer cells, human brain tissue and mice. Although H3K27ac and

---

**Fig. 1 | H4K16ac and H3K122ac are enriched at the 5′ UTR of L1 and ERV/LTRs in hESCs. a**, Bar chart showing the percentage distribution (*y* axis) of histone PTMs CUT&Tag peaks across ChromHMM chromatin features. Low signal (Lo), transcription (Txn) **b**, Dot plot showing the ratio (observed/expected) of enrichment of CUT&Tag peaks across gene transcription start sites, TE families (L1, ERVLTRs, SINE/Alu-Alu family of short interspersed nuclear elements) and the gene body. The circle size represents the log₂ value for the ratio, and the color range represents the enrichment ratio. **c**, Percentage distribution of repeat elements: Alu, ERV_classI & ERV_classII (endogenous retrovirus class I & II), ERVL_MaLRs (endogenous retrovirus type-L mammalian apparent retrotransposon, hAT_Charlie (member of hAT superfamily of DNA transposon), L3/CR1 (long interspersed nuclear elements 3/chicken repeat1), LINE1, LINE2

(long interspersed nuclear elements 2), MIRs (mammalian inverted repeats) and TcMar-Tigger (TcMar-Tigger DNA transposon, Tigger2 subfamily) for CUT&Tag peaks. **d**, Illustration showing the structure of human L1s (above), two open reading frames (ORF1 and ORF2), along with endonuclease (EN), reverse transcriptase (RT) and carboxyl terminal segment (C) within the ORF2 are shown. Heatmap displaying the histone modification CUT&Tag signal (counts per million, CPM) at 10,538 (n) full-length L1s (>5 kb, left) and NCBI Ref-seq genes (right); data for three replicates, R1, R2, and R3, are plotted separately. **e**, UCSC Genome Browser tracks (Hg38) showing signal density (CPM) of histone modifications (individual replicates) at L1PA10, a representative L1 subfamily, (left) and the *L1PA4*, *ERV1*, and *USP38* genes (right).

H4K12ac were detected at some L1 5′ UTRs, they were enriched at a much higher level at the promoters of genes (Fig. 1b). Interestingly, along with H4K16ac and H3K122ac, L1 5′ UTRs were also enriched with H3K4me1 but were depleted of H3K4me3 (Fig. 1d), suggesting that these elements could function as CREs.

### H4K16ac⁺ L1 5′ UTRs are enriched with enhancer features

LTR subfamilies function as enhancers to regulate genes in a tissue-specific manner in humans and mice (reviewed in ref. 16). LTR5- and LTR7-related subfamilies function as enhancers in hESCs[21,43,44]. However, whether L1 elements can act as enhancers to regulate genes

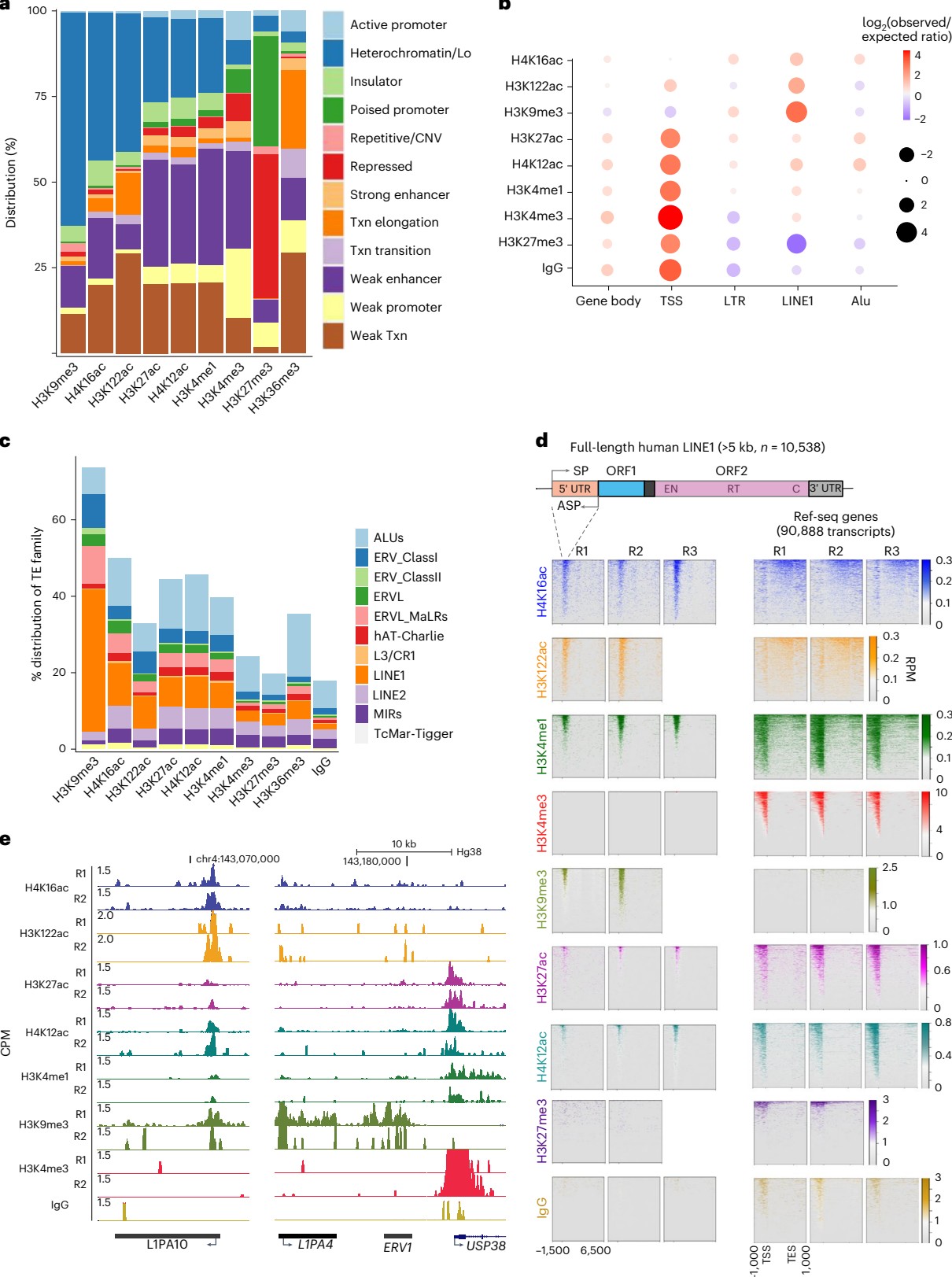

in *cis* is not known. Here we found that H4K16ac is particularly enriched at the 5′ UTR of full-length L1 subfamilies and correlates with chromatin features associated with active enhancers, such as H3K27ac, H3K4me1, BRD4 and ATAC-seq signal (Fig. 2a,d).

Interestingly, not all L1 subfamilies are enriched with active enhancer features at the same level. The evolutionarily younger L1s (L1HS, L1PA2 and L1PA3, 3–12.5 million years) are enriched with active enhancer features, including H4K16ac. These L1s are known to be transcriptionally active. Despite evolutionarily older L1s being transcriptionally inactive, the 5′ UTRs of these L1 subfamilies (L1PA7–L1PA16, 31–80 million years) are enriched with H4K16ac, along with other active enhancer features, but H3K9me3 is less enriched (Fig. 2a), suggesting that the 5′ UTRs of older full-length L1s have been co-opted to function as functional regulatory elements.

Analysis of genome-wide enhancer activity data (using self-transcribing active regulatory region sequencing, or STARR-seq), generated by ENCODE[45] from neuroblastoma (SH-SY5Y) and erythroleukemia (K562) cell lines, showed enhancer activity specifically at the 5′ UTR of L1s (Fig. 2b) in a cell-type-specific manner. The presence of active enhancer chromatin features (Fig. 2a) and the ability of L1 5′ UTR to drive transcription of the minimal promoter in an in vitro enhancer reporter assay (Fig. 2b) further confirmed that 5′ UTRs of full-length L1s could function as transcriptional enhancers.

## LTRs with H4K16ac process higher enhancer activity

Our data show that, apart from the LTR5 and LTR7 elements that show clear enrichment of active enhancer chromatin features, some of the subfamilies of LTR16 and LTR33 may also serve as enhancers in hESCs, because they are enriched with H4K16ac and other active enhancer chromatin features (Fig. 2c,d). Interestingly, analysis of STARR-seq data from K562 and SH-SY5Y cells revealed that H4K16ac+ LTRs in these cell lines show significantly higher enhancer activity than LTRs that overlap with only H3K4me1 or H3K27ac peaks (Fig. 2e,f and Extended Data Fig. 4a). These results further support the notion that H4K16ac+ LTRs function as enhancers. The rest of the LTR and Alu families are not likely to act as enhancers in hESCs, as they lack known enhancer chromatin features (Extended Data Fig. 4b).

## TEs marked with H4K16ac are bound by looping factors

We aimed to identify TFs bound at H4K16ac+, H3K27ac+ and H3K122ac+ TEs using TF ChIP–seq data from ENCODE. Expectedly, EP300 is enriched at LTRs marked with H3K27ac (Fig. 3a). YY1 is enriched at L1s marked with all three marks, supporting the known role of YY1 in activating L1 transcription[46]. CTCF and RAD21 showed higher enrichment at H4K16ac+ and H3K122ac+ L1s and LTRs than at H3K27ac+ L1s and LTRs. MYC and KDM1A were depleted at H4K16ac+ and H3K27ac+ L1s. These observations are consistent with previous reports showing the role of CTCF and RAD21 in activating L1 transcription[47,48], and of MYC and KDM1A in repressing L1 transcription[49]. SP1, TCF12 and NANOG binding was also specifically enriched at H3K27ac+ L1 and LTRs, suggesting that they have a role in transcription at these elements.

## L1s and LTRs with acetylation marks loop with genes

YY1, enriched at acetylated L1s, functions as a looping factor that facilitates interaction between enhancers and promoters[50]. Compared with L1s with acetylation marks, LTRs and Alu elements marked with histone acetylations are enriched with USF1, REST and a looping factor ZNF143 (Fig. 3a and Extended Data Fig. 5). Meta-analysis confirmed the enrichment of CTCF, RAD21 and YY1 at both H3K27ac+ and H4K16ac+ L1 5′ UTRs and LTRs (Fig. 3b). Analysis of published Hi-C data revealed that, compared with TEs that lack acetylation marks, TEs with such marks are enriched at topologically associated domain (TAD) borders (Fig. 3c). Moreover, H4K16ac levels are relatively higher at TEs overlapping with the TAD borders than at TEs that do not overlap with TAD borders (Fig. 3d,e). Furthermore, to identify whether TEs with histone acetylations loop with genes, we called significant loops from publicly available micro-C data from H1 hESCs[51]. This revealed that the fraction of TEs (L1, LTRs and Alu elements) with acetylation marks that form chromatin loops with genes is significantly higher than the fraction of TEs lacking these marks (Fig. 3f,g). These analyses provide evidence that transcribed TEs enriched with histone acetylation marks could contribute to three-dimensional (3D) chromatin folding and looping interactions with genes.

## H4K16ac+ LTR/HERVs act as enhancers

We aimed to use CRISPR interference (CRISPRi) to investigate the role of H4K16ac+ TEs in regulating genes in *cis*, by recruiting KRAB repressor domain (dCAS9-KRAB) to TEs. We performed CRISPRi for individual LTRs of human endogenous retrovirus (HERV) or L1 5′ UTRs by co-transfecting two independent guide RNAs that recruit dCAS9-KRAB to specific TEs enriched with H4K16ac in hESCs. We then performed quantitative reverse transcription PCR (RT–qPCR) for nearby expressed genes or genes that show the looping interaction in the RAD21-HiChIP data (Fig. 4a)[52]. CRISPRi targeting an H4K16ac+ LTR7/HERV-H (HERV type H family) element located ~50 kb away from *PEX1* and ~30 kb from the *GATAD1* promoter led to downregulation of *PEX1*, but not *GATAD1* (Fig. 4b,d). CRISPRi targeting another H4K16ac+ LTR7/HERV-H element located ~50 kb away from the *NUS1* promoter led to the downregulation of the *NUS1*, but not of *GOPC* (Fig. 4c,d). However, CRISPRi targeting H4K16ac+ LTRs/HERV-L-18 (HERV type L-18 family) and HERV-L-18 int (internal portion of HERV-L-18) that are close to TAD borders (Figs. 3d and 4d) did not show downregulation of nearby genes *ZC3H15* and *ODF2L*, suggesting that some but not all H4K16ac+ HERV/LTR loci function as enhancers. However, it is possible that such H4K16ac+ TEs could contribute to LTR/HERV transcription and 3D genome folding (Fig. 3c–e)[53].

## H4K16ac+ L1 5′ UTRs function as enhancers

We next focused on L1s and asked whether L1 5′ UTRs enriched with H4K16ac regulate genes in *cis* by performing CRISPRi for H4K16ac+ L1 5′ UTRs, together with two L1 5′ UTRs that lack detectable histone acetylation marks. CRISPRi for the H4K16ac+ 5′ UTR of an L1PA10 located ~110 kb upstream of *USP38* led to specific downregulation of *USP38*

**Fig. 2 | H4K16ac+ TEs are enriched with chromatin features associated with enhancer activity. a**, Heatmap of CUT&Tag signals for histone modifications and BRD4 ($n$ = 2 or 3 biological replicates), normalized to IgG and ATAC-seq signal at TE subfamilies; −1.5 kb to +6.5 kb from the full-length L1 start sites (>5 kb). **b**, Heatmap showing H4K16ac and H3K27ac CUT&Tag and STARR-seq signal, normalized to input, in K562 and SH-SY5Y cells. **c**, Like **a**, but for ±2.5 kb around the ERV/LTR center for subfamilies of LTR5, LTR7, LTR9, LTR16 and LTR33. The number of LTRs in each subfamily are shown below. Data for the Alu subfamily and the rest of the LTR subfamilies are in Extended Data Figure 5. **d**, Genome browser tracks (Hg38) showing the average ($n$ = 2 or 3 biological replicates) CPM for two replicates of H4K16ac, H3K122ac, H3K27ac, H3K4me1 and H3K4me3 CUT&Tag data from hESCs. RepeatMasker tracks showing L1 (L1PA7, top), LTR5, LTR16 and LTR33 (bottom), and ENCODE-layered H3K27ac and CREs are shown below each panel. **e**, Violin plots showing STARR-seq signal from K562 cells ($n$ = 2 biological replicates, signal normalized to input) across LTRs intersecting H3K4me1 peaks; H4K16ac but not H3K4me1 peaks; H3K4me1 and H4K16ac peaks; and H3K4me1 and both H3K27ac and H4K16ac peaks. **f**, Like **e**, but for SH-SY5Y cells ($n$ = 2 biological replicates, signal normalized to control) across LTRs that intersect with no H4K16ac or H3K27ac peaks ($n$ = 40,000 LTRs); H3K27ac but not H4K16ac peaks ($n$ = 22,447 LTRs); H4K16ac but not H3K27ac peaks ($n$ = 35,349 LTRs); and H4K16ac and H3K27ac peaks ($n$ = 15,602 LTRs). In all box plots, center lines indicate the median, bounds indicate the 25th and 75th percentiles, and whisker limits show 1.5 × interquartile range; $P$ values for all the violin and box plots were calculated using the pairwise two-sided multi-comparison Dunn test for post hoc testing, following the Kruskal–Wallis test with Bonferroni correction.

but not other nearby genes *GAB1* and *SMARCA5*. Notably, CRISPRi for two H4K16ac[−] 5′ UTRs of L1s, located ~30 kb and ~85 kb from the *USP38* promoter, led to no change in the USP38 transcript level, showing the specificity of the H4K16ac[+] L1PA10 element in regulating *USP38* (Fig. 4e,j). Similarly, CRISPRi for the H4K16ac[+] 5′ UTR of L1PA10, located ~270 kb from the *TANC2* promoter, led to downregulation of *TANC2*,

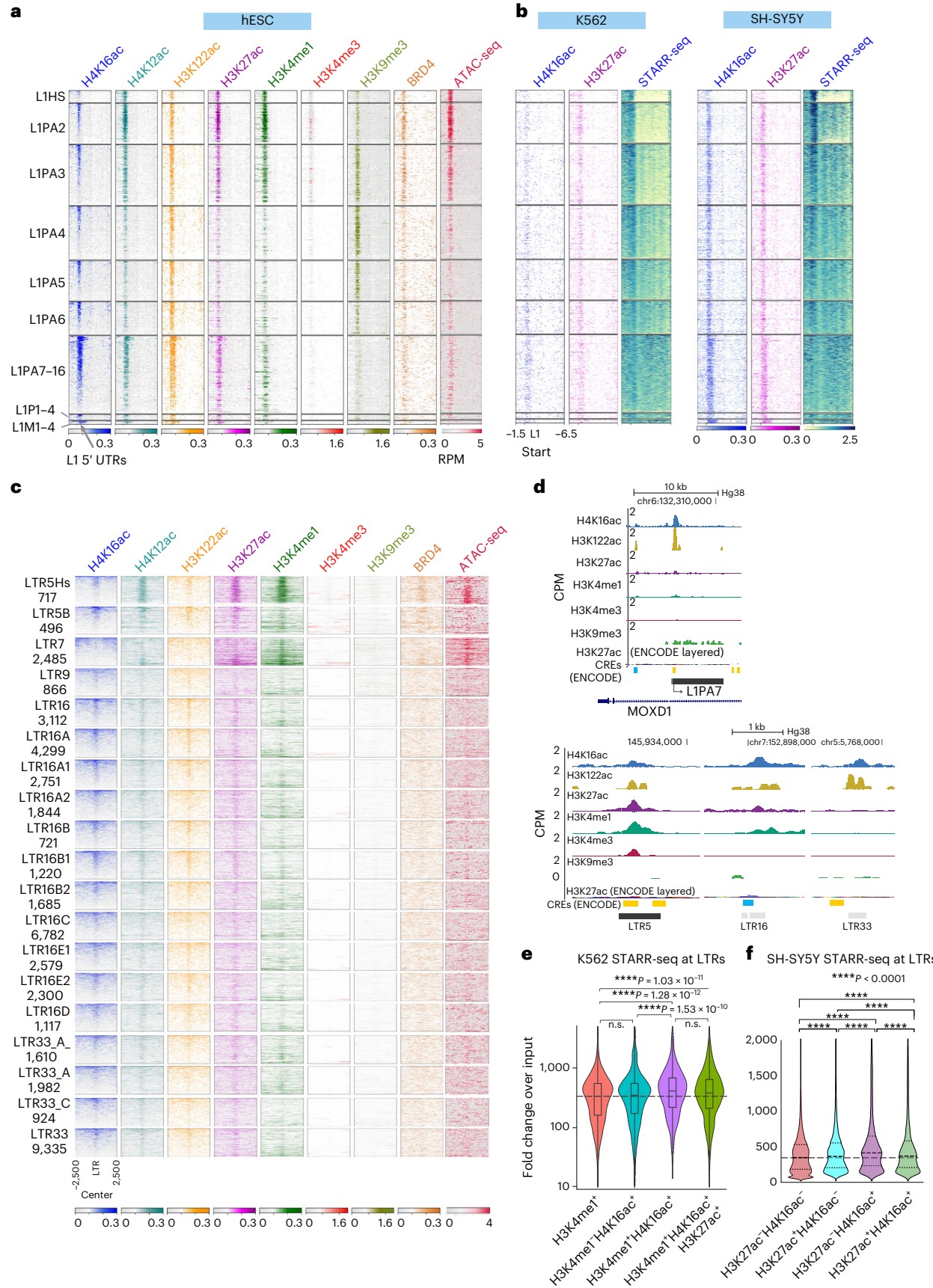

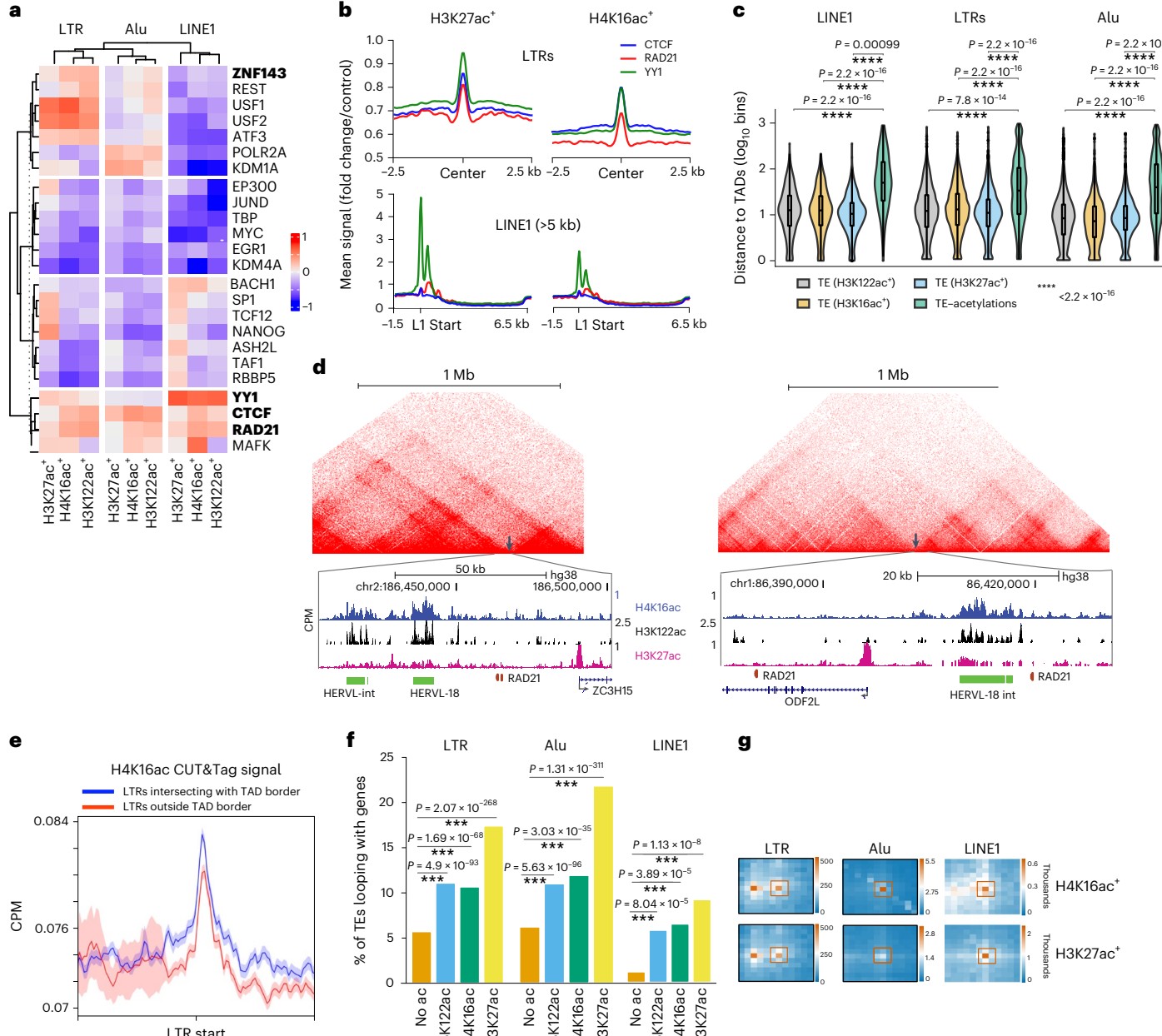

**Fig. 3 | H4K16ac⁺ L1 and LTRs are enriched at TAD borders and loops with genes. a**, Heatmap shows the difference/sum (details in Methods) ratio for observed and expected occurrences of TF-binding sites in H4K16ac, H3K27ac and H3K122ac peaks at the 5′ UTR of L1, ERV/LTR or SINE/Alu, over the random background. Looping factors that are known to be enriched at enhancer-promoter loops are in bold; a complete list of TFs is in Extended Data Figure 6. **b**, Average type summary plots showing the mean signal distribution (fold change/control) of YY1 (green), RAD21 (red), and CTCF (blue) at LTRs (top) and full-length L1 (>5 kb, bottom) that overlaps with H3K27ac (left) or H4K16ac (right). **c**, Violin plot showing the distance to TAD borders (y axis, log₁₀ bins) for LTR (H4K16ac⁺ #10258, H3K27ac⁺ #8063, H3K122ac⁺ #17132), Alu element (H4K16ac⁺ #7394, H3K27ac⁺ #4659, H3K122ac⁺ #61312) and L1 (H4K16ac⁺ #892, H3K27ac⁺ #550, H3K122ac⁺ #1439) marked with H3K27ac, H4K16ac or H3K122ac, and for TEs that lack these marks (LTR #31678, Alu #31589 and L1 #452). P values

for the violin plots were calculated by Mann–Whitney U test. **d**, Example UCSC Genome Browser tracks showing H4K16ac, H3K122ac and H3K27ac signals at TAD borders (arrow marks) (micro-C data from H9 hESC). CRISPRi was used for some of these HERV/LTRs for validation (Fig. 4d). **e**, Average type summary plot depicting IgG-normalized H4K16ac signal (CPM), with standard error (shaded area), at the LTRs overlapping the TAD border (blue) and LTRs elsewhere in the genome (red). **f**, Bar graph showing the percentage of H4K16ac⁺, H3K27ac⁺ and H3K122ac⁺ TEs and TEs that lack these marks (full-length L1, LTR and Alu) that contact genes through chromatin loops (P values were calculated by Fisher's exact test). (# same as in **c**). **g**, Aggregate peak analysis (APA) plots for H4K16ac⁺ and H3K27ac⁺ LTRs, Alu elements, and L1s that contact genes through loops. The number of contacts (in thousands) are shown in the scale bars. In all box plots, center lines indicate the median, bounds indicate the 25th and 75th percentiles, and whisker limits show 1.5 × interquartile range.

but not *CYB561* (Fig. 4f,j). CRISPRi for H4K16ac⁺ L1PA7, located ~24 kb from the *COMMD10* promoter, also led to a significant downregulation of *COMMD10*, but not the nearby gene *SEM6A* (Fig. 4g,j). RAD21-HiChIP data and the micro-C analysis revealed significant looping interactions

between *MOXD1* and *STX7* genes with the H4K16ac⁺ L1PA8, located ~100 kb away from the *MOXD1* promoter (Fig. 4h,j). CRISPRi for the 5′ UTR of this L1PA8 led to significant downregulation of both *MOXD1* and *STX7*. However, the expression of *ENPP1*, which does not loop

with this L1, was not altered (Fig. 4h,j), demonstrating the specific *cis*-regulatory function of these L1s.

To further confirm that H4K16ac⁺ L1 5′ UTRs regulate genes in *cis*, we used CRISPR–CAS9 to delete full-length L1 elements in H1 hESCs. Owing to the difficulty of specific deletion of L1 5′ UTRs, we nucleofected the cells with pairs of synthetic guide RNAs along with CAS9 (ribonucleocomplex) that target the flanking region of four full-length L1s (~7 kb deletions). We generated two independent clonal lines with heterozygous deletions for L1PA10 and one clone for L1PA7; both are H4K16ac⁺ and are located upstream of *USP38* (Fig. 4e and Extended Data Fig. 6a,b). In accordance with CRISPRi data, RT–qPCR data showed that the deletion of L1PA10 and L1PA7 led to the downregulation of *USP38*, but not other nearby genes that were tested, namely *GAB1* and *SMARCA5* (Fig. 4k). For deletion of L1s located at the *MOXD1* and *RLN2* loci (Fig. 4g,i), we nucleofected gRNA–CAS9 ribonucleoprotein complexes and used two independent pools of hESCs that showed ~50% deletion efficiency (Extended Data Fig. 6c). Although CRISPRi for L1PA8 resulted in the downregulation of both *MOXD1* and *STX7*, genetic deletion led to specific downregulation of *MOXD1*, but not *STX7* and *ENPP1* (Fig. 4h,k). Deletion of another H4K16ac⁺ L1PA7 located downstream of *RLN2*, ~12 kb away from the *RLN2* promoter, led to the downregulation of *RLN2* but not a nearby gene *PLGRKT* (Fig. 4i,k). Overall, CRISPRi and genetic deletion experiments confirmed that H4K16ac⁺ L1s and LTRs are involved in regulation of transcription of genes in *cis*.

### MSL and H4K16ac activate transcription of TEs

Next, we aimed to deplete H4K16ac to investigate whether it regulates TE transcription. H4K16ac is catalyzed explicitly by KAT8 when associated with the MSL complex, but not the NSL complex[32–35] (Fig. 5a). Because depletion of the individual MSL complex proteins MSL1, MSL2 and MSL3 is sufficient to reduce H4K16ac level[54], we knocked down MSL3 using two independent lentiviral small hairpin RNAs (shRNAs) in H9 hESCs; we first validated the depletion by RT–qPCR, which showed ~50% downregulation of MSL3. RT–qPCR with primers recognizing full-length L1 subfamilies, such as human-specific (L1HS), mammalian-wide (L1M) and primate-specific (L1PA and L1PB) full-length L1s, showed significant downregulation upon MSL3 knockdown (KD). Similarly, RT–qPCR with primers recognizing HERV-K and HERV-H transcripts showed significant downregulation of HERV-H and HERV-K in MSL3-depleted hESCs (Fig. 5b). Western blotting confirmed that MSL3 depletion led to a specific reduction in H4K16ac but not H3K27ac (Fig. 5c and Supplementary Fig. 1). Like the transcript data, L1-ORF1 protein (L1-ORF1p), encoded by full-length L1s (Fig. 1d) and HERV envelope protein (antibody raised against ERVW-1) were also reduced upon MSL3 and H4K16ac depletion (Fig. 5c), consistent with the high level of H4K16ac at L1 5′ UTRs (Fig. 1d) and ERVW-1 locus (Fig. 4c).

We further used doxycycline-inducible Cas9 (iCAS9)-mediated knockout (KO) of MSL1 in H1 hESCs (Fig. 5d) and in TDFs (Extended Data Fig. 7) to confirm our findings from the shRNA-mediated MSL3 depletion. Immunofluorescence for H4K16ac and L1-ORF1 protein followed by high-content imaging revealed a significantly reduced number of

L1-ORF1p foci in H4K16ac-depleted MSL1-KO hESCs (Fig. 5d). Like hESC data, MSL3 KO in TDFs reduced the bulk of H4K16ac (Extended Data Fig. 7a) and at L1 5′ UTRs and LTRs (Extended Data Fig. 7d).

RNA-seq data analysis from MSL3-KO TDFs showed significant downregulation of L1 and LTR transcripts (Extended Data Fig. 7b,c). Notably, H4K16ac⁺ L1s, but not H4K16ac⁻ L1s, are significantly downregulated in MSL1-KO TDFs (Extended Data Fig. 7b). All these results confirm the direct role of MSL mediated H4K16ac in the transcriptional activation of L1. MSL3-KD RNA-seq analysis in hESCs showed that pluripotency and differentiation-associated genes were unaffected (Extended Data Fig. 8a,b). However, H4K16ac⁺ genes were more affected than were H4K16ac⁻ genes (Extended Data Fig. 8c). Further analysis of L1s and LTRs showed significant downregulation of both human-specific (L1HS) and primate-specific (L1PA2 to L1PA16) full-length L1 and LTR subfamily transcripts (Fig. 5b,g). L1, LTRs, HERV-K and HERV-L transcripts and protein-coding genes also show small but significant downregulation in MSL3-KD cells (Fig. 5e,f and Extended Data Fig. 8d).

### MSL/H4K16ac at TEs maintain active *cis*-regulatory landscape

H4K16ac causes chromatin decompaction in vitro, and depletion of H4K16ac has been shown to reduce chromatin accessibility[29,55]. Therefore, we asked whether the lack of H4K16ac leads to altered accessibility at TEs. ATAC-seq data showed a specific reduction in accessible DNA at the 5′ UTR of L1s in MSL3-depleted hESCs (Fig. 5g). In particular, evolutionarily younger L1s show a decrease in DNA accessibility, accompanied by reduced transcriptional activity at these elements (Fig. 5g).

Genes closer to H4K16ac⁺ L1 and H4K16ac⁺ LTRs are significantly highly expressed compared with genes farther away from these L1s and LTRs. By contrast, genes closer to H4K16ac⁻ L1 and H4K16ac⁻ LTRs show significantly lower expression levels than farther genes (Fig. 6a,b). Next, we asked whether depletion of MSL/H4K16ac at L1 and LTRs affects the expression of genes located near these TEs marked with H4K16ac⁺ TEs. MSL3 depletion led to a small but significant downregulation of many transcripts (*n* = 3,312) closer (<10 kb) to H4K16ac⁺ L1s (Fig. 6c). Similarly, many transcripts that are closer (<10 and <25 kb) to H4K16ac⁺ LTRs are significantly downregulated compared to transcripts that are 25 to 50 kb away (Fig. 6d).

Overall, our results confirm the role of MSL/H4K16ac at L1 and LTRs in transcriptional activation of TEs (Fig. 5b,d–f) and in regulating genes that they are associated in linear distance or 3D space (Figs. 4 and 6a–e). Therefore, we conclude that MSL complex-mediated acetylation of H4K16 leads to the opening of chromatin structure and increased transcriptional activity at L1 and LTRs in a cell-type-specific manner. The permissive local chromatin environment at H4K16ac⁺ TEs shapes the *cis*-regulatory landscape across the mammalian genome (Fig. 6e).

## Discussion

TEs are repressed by many epigenetic pathways, such as DNA methylation, H3K9me3, KRAB-ZNF, HUSH complex and piwi-interacting RNA (piRNAs). We have discovered that the MSL-H4K16ac axis functions as

---

**Fig. 4 | H4K16ac⁺ L1 5′ UTRs function as enhancers to regulate genes in *cis*.**
**a**, Illustration showing CRISPRi and CRISPR-mediated deletion strategy for TEs. Genes that show looping interaction (in RAD21-HiChIP data) and that are expressed in hESCs were chosen as putative targets for RT–qPCR, and other nearby expressed genes were chosen as controls. **b,c**, Genome browser tracks showing H4K16ac and H3K27ac CUT&Tag data (CPM) at LTR7/HERV-H-int and LTR/ERV1 loci and their putative target genes. **d**, RT–qPCR data showing relative fold change (normalized to *ACTB*) in the expression of putative target genes *NUS1* and *PEX1* upon CRISPRi for HERV/LTRs, but not other nearby genes (*GOPC* and *GATAD1*). **e–i**, Like **b** and **c**, the genome browser track shows CUT&Tag data at L1PA10 at the *TANC2* locus, L1PA7 at the *COMMD10* locus, L1PA7 at the *MOXD1* locus, L1PA10 and L1PA7 at the *USP38* locus, and L1PA7 at the *RLN2* locus. L1PA2 and L1MA27 at the *USP38* locus, which lack histone acetylation marks, were used

as controls. **j**, Same as **d**, but for H4K16ac⁺ or H4K16ac⁻ putative target genes for L1s (L1PA2 and L1MA2). *TANC2*, *COMMD10*, *MOXD1*, *STX7*, and *USP38* were selected as putative target genes, along with *CYB561*, *SEMA10A*, *ENPP1*, *GAB1*, and *SMARCA5* were selected as putative non-targets. **k**, Same as **j**, but RT–qPCR was done upon CRISPR–CAS9-mediated deletion of full-length L1. Two independent clones for L1PA10 (H4K16ac⁺) and one for L1PA7 (H4K16ac⁺), L1PA2 (H4K16ac⁻) and L1MA2 (H4K16ac⁻) located at the upstream of *USP38* were tested, and for L1PA7 located at *MOXD1* and *RLN2*, the pools of cells were tested. For all RT–qPCR experiments, data are shown as mean ± s.d. from *n* = 3 independent experiments; *P* values are from unpaired *t*-test with Welch correction; the two-stage step-up (Benjamini, Krieger and Yekutieli) method was used, and the false-discovery rate (FDR) was 1.00% for multiple comparisons. n.s., not significant.

a transcriptional activator of TEs, particularly L1s and LTRs. TEs have contributed substantially to the evolution of mammalian genomes by helping to shape both the coding and noncoding regulatory landscape. Several ERV/LTR subfamilies have been demonstrated to function as

tissue-specific enhancers. Here, we have demonstrated that L1 5′ UTRs and LTR/ERVs enriched with acetylated histones loop with genes, and L1s and LTRs marked with H4K16ac function as enhancers to regulate genes in *cis*.

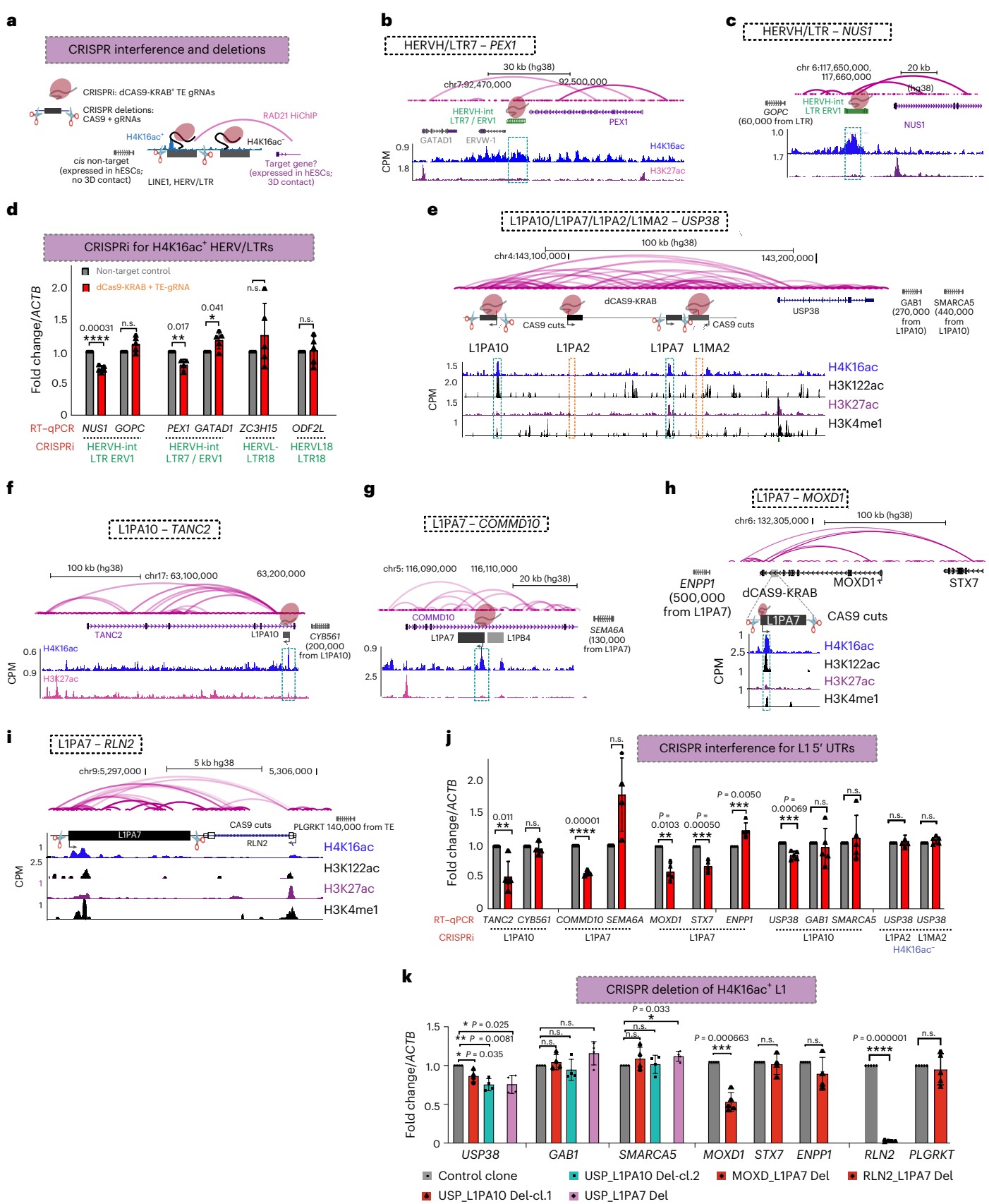

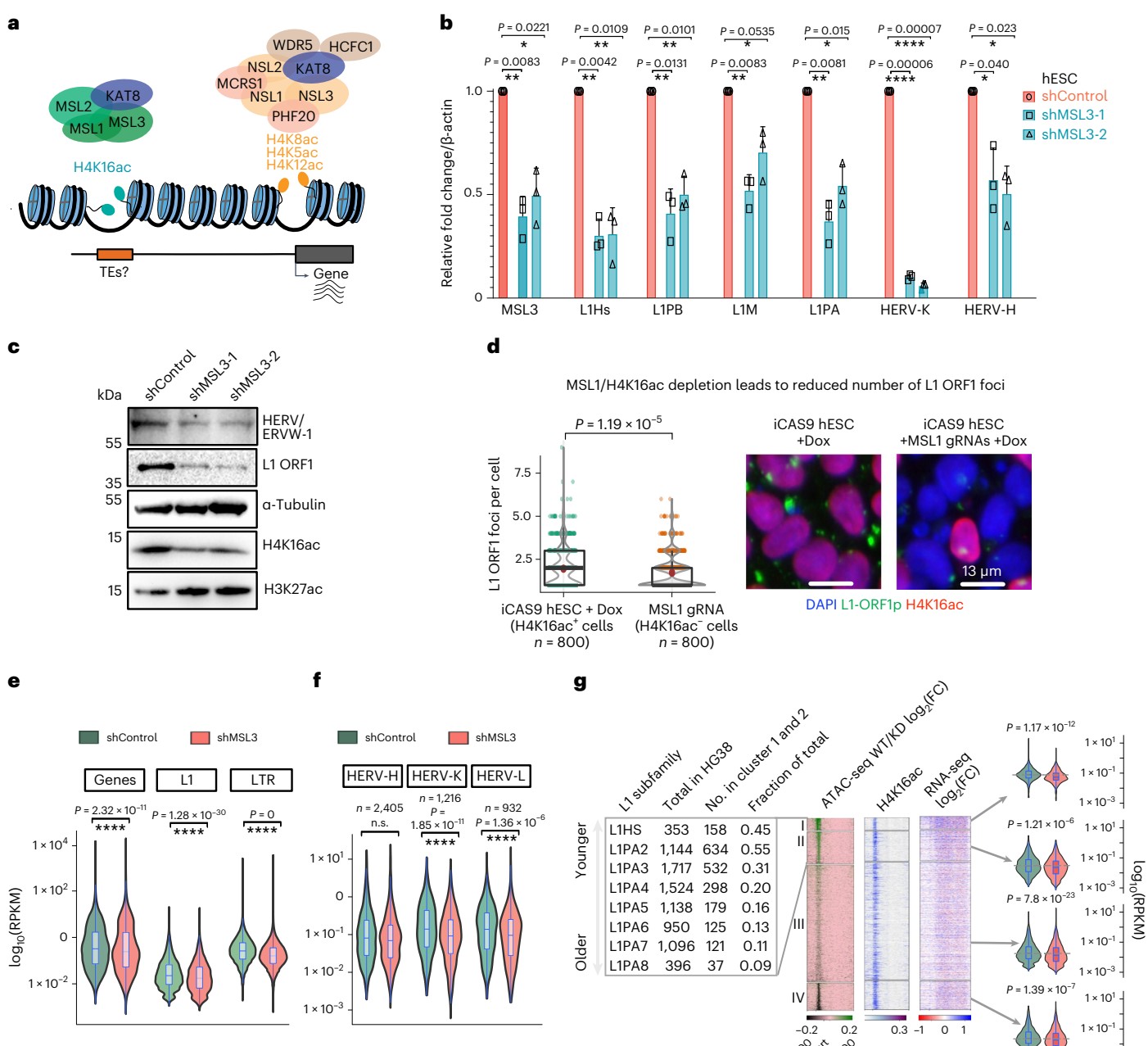

**Fig. 5 | MSL activates transcription of TEs. a**, Illustration showing that KAT8 catalyzes H4K16ac only when bound to the MSL complex, not the NSL complex. **b**, RT–qPCR data from hESCs showing mean fold change (normalized to β-actin) in MSL3, L1 and HERV subfamilies upon lentiviral shRNA-mediated KD of MSL3 using two independent shRNAs, versus hESCs transfected with a non-targeted control shRNA. Data are shown as mean ± s.d. from $n = 3$ independent experiments; $P$ values were calculated using an unpaired $t$-test with Welch correction; the two-stage step-up (Benjamini, Krieger and Yekutieli) method was used, and the FDR was 1% for multiple comparisons. **c**, Western blots showing HERV, L1-ORF1, and H4K16ac levels after shRNA-mediated knockdown of MSL3 described in **b**; α-tubulin and H3K27ac served as controls in control and MSL3-KD hESCs (data are representative of $n = 2$ independent experiments; uncropped images are in Supplementary Data Fig. 1). **d**, Representative images (right) and quantification of high-content (automated microscopy) imaging data (left) showing the number of L1 ORF1p foci per cell in H4K16ac⁺ and H4K16ac⁻ MSL1-KO cells. Eight hundred cells per condition were analyzed in two

wells. Data are representative of $n = 2$ independent experiments; $P$ values were calculated using Welch's $t$-test with 95% confidence interval. Scale bar, 13 μm. **e**, Violin plots showing RNA-seq for genes, full-length L1s, and LTRs for control and MSL3-KD hESCs ($n = 4$). **f**, RNA-seq signal at HERV subclasses HERV-K, HERV-H, and HERV-L. **g**, Left, heatmaps showing H4K16ac (CPM), ATAC-seq (CPM) and RNA-seq (log₂(fold change)) for control/MSL3 KD in hESCs, $n = 4$) across full-length L1 with $K$-means clusters. The distribution of L1 subfamily members in clusters 1 and 2 is shown on the left; multi-mapped reads were retained for these heatmaps. Right, violin plots showing RNA-seq signal (log₁₀(reads per kilobase of transcript, per million mapped reads, RPKM), control and MSL3-KD hESCs) at four L1 clusters. In all box plots, center lines indicate the median, bounds indicate the 25th and 75th percentiles and whisker limits show 1.5 × interquartile range. $P$ values for all the violin and box plots were calculated using the pairwise two-sided multi-comparison Dunn test, used for post-hoc testing following the Kruskal–Wallis test, with Bonferroni correction.

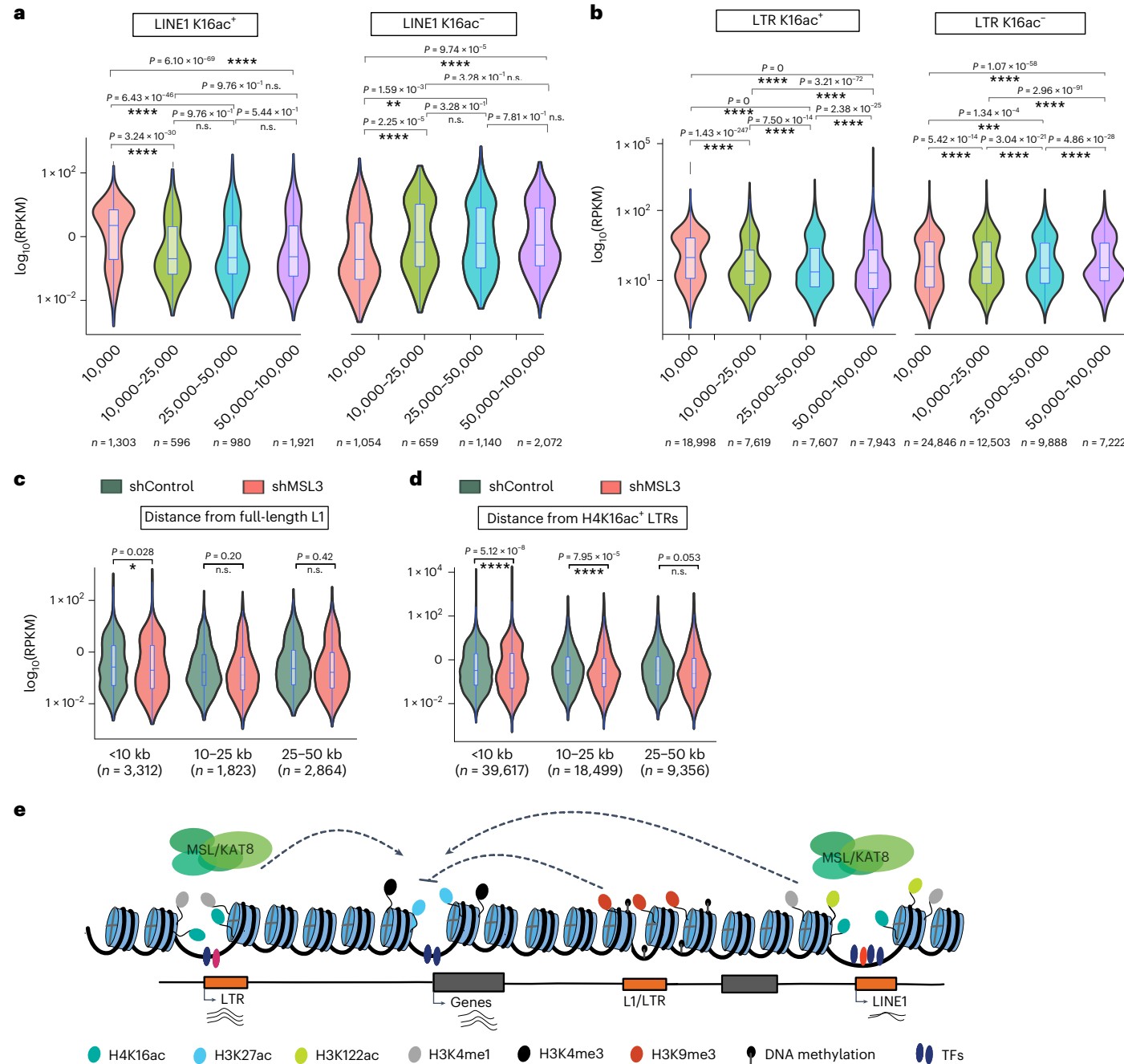

**Fig. 6 | H4K16ac maintains an active chromatin landscape at TEs. a**, Violin plots showing the distance-dependent effect on the expression of genes from L1 with H4K16ac peaks (H4K16ac⁺) (left). Genes close to L1 that lack detectable H4K16ac peaks (H4K16ac⁻) are in TDF cells (right). X–axis shows the distance from the TEs. **b**, Like **a**, but for LTRs. **c**, hESC RNA-seq signals for control (shControl) and MSL3 knockdown (shMSL3) at genes that lie 10 kb, 10–25 kb, or 25–50 kb away from the H4K16ac-overlapping full-length L1s. **d**, Like **c**, RNA-seq signals at genes that lie in 10 kb, 10–25 kb or 25–50 kb away from the H4K16ac-overlapping LTRs. **e**, The working model shows MSL/KAT8-mediated H4K16ac maintains accessible chromatin, activates transcription at TEs, and contributes to their enhancer activity to regulate genes in *cis*. In all box plots, center lines indicate the median, bounds indicate the 25th and 75th percentiles, and whisker limits show 1.5 × interquartile range. *P* values for all the violin and box plots were calculated using the pairwise two-sided multi-comparison Dunn test, used for post hoc testing following the Kruskal–Wallis test with Bonferroni correction.

Roadmap epigenomics data have shown that TEs are depleted of H3K27ac and accessible chromatin; only 3% of TE bases are annotated with active regulatory chromHMM states, compared with 32% of promoter bases[56]. Despite that, TEs contribute up to 40% of TF-binding sites; hence, TEs have been proposed to contribute to species- and tissue-specific rewiring of gene regulatory networks[57–59]. This suggests that an unknown chromatin pathway could contribute to the enhancer activity of TEs in a cell-type- or species-specific manner, which could

be independent of H3K27ac. Our CUT&Tag data show that the level of H3K27ac is much higher at genes and promoters than at TEs. However, H4K16ac is enriched explicitly at the L15′ UTRs, along with several other chromatin features associated with active enhancers. We now demonstrate that L15′ UTRs marked with H4K16ac along or together with H3K122ac and H3K27ac function as enhancers to regulate the expression of genes in *cis*. Although L1s are expressed at higher levels during early development, including stem cells, they are also upregulated in

cancer and the neuronal lineage. Consistently, we found that H4K16ac is enriched at L1 5′ UTRs in human and mouse stem cells, cancer cell lines, and post mortem brain tissues, suggesting that 5′ UTRs of L1s bound by tissue-specific TFs and enriched with histone acetylations could function as tissue-specific enhancers. Enrichment of H4K16ac at TEs, which constitute a major part of the mammalian genome, is consistent with findings showing that nearly 30% of the histone H4 is acetylated at H4K16 (ref. 34).

Many LTR subfamilies are enriched with active enhancer associated chromatin features indicating that they could function as active enhancers. It has been proposed that some of the LTR subfamilies are essential in driving the expression of lineage-specific genes[43,57]. However, only a minority of putative RLTR13D6 subfamily-derived enhancers identified through epigenomic analyzes have been experimentally validated to function as enhancers[17]. This highlights the importance of functional validations using CRISPR-based perturbation of candidate TEs enriched with enhancer chromatin features. Although we found all of the tested CRISPR-edited H4K16ac⁺ L1s downregulated their putative target genes in *cis*. Genome-wide enhancer reporter assays, in combination with systematic genome-scale perturbation, are needed to identify what fraction of L1s and LTRs with H4K16ac function as enhancers.

TEs with acetylation marks, including H4K16ac, are bound by looping factors, including CTCF, RAD21, YY1, and ZNF143. Moreover, the fraction of these TEs that loop with genes is significantly larger than the fraction of TEs without acetylation marks that do so, further supporting the role of transcriptionally active TEs in rewiring the regulatory landscape in a species- and cell-type-specific manner[53]. Since our results show that the MSL-H4K16ac axis drives transcription at TEs, including HERVs (Fig. 5b,f and Extended Data Fig. 4b), we hypothesize that MSL-H4K16ac-mediated transcription at TEs likely contributes to the rewiring of 3D chromatin organization at transcriptionally active TEs, as RNA polymerase II transcription drives enhancer-promoter contact[60]. The factors contributing to the recruitment of the MSL complex to the specific genomic region are unknown in mammals. Intriguingly, the role of MSL complex in co-opting TEs to rewire *cis*-regulatory elements appears to have been conserved during the evolution of dosage compensation in *Drosophila miranda*, in which a mutant helitron TE has been shown to recruit the MSL complex to the evolutionarily young X chromosome to increase transcription[61]. In *Drosophila* dosage compensation, expression of most X-linked genes is increased approximately twofold by H4K16ac, specifically in males[62]. This MSL-mediated X upregulation appears to be conserved in mammals, in which H4K16ac has been shown to upregulate genes on the single active X chromosome to balance expression with two copies of the autosomes[63]. Interestingly, X chromosomes have a higher number of L1s than autosomes[64], suggesting that MSL-H4K16ac at L1s in the X chromosome could contribute to X upregulation.

TFs enriched at H4K16ac⁺ TEs (Fig. 3a and Extended Data Fig. 5) could contribute to maintaining MSL-H4K16ac and transcription at TEs. Notably, the MSL complex recruits YY1 to the *Tsix* promoter to activate its expression in mESCs[32], suggesting a possible interplay between MSL and YY1 in regulating L1 transcription. Interestingly, MAFK, which has previously been reported to be enriched at TEs[59], is enriched explicitly at H4K16ac⁺ L1 5′ UTRs, suggesting a potential interplay between MAFK and MSL complex.

Neuronal cells have high L1 expression and retrotransposition[65]; retrotransposon dysregulation is also linked with neurological disorders[1]. TEs and their transcriptional regulators play wider roles in shaping transcriptional networks during early human development[66]. Loss of function mutations in genes encoding KAT8 containing protein complexes such as *KANSL1, MSL3* and *KAT8* lead to neurodevelopmental disorders[37,67–69]. Enrichment of H4K16ac at the 5′ UTRs of L1s in human brain tissues suggests that altered gene expression programme due to TE dysregulation in the nervous system could be a possible mechanism for these disorders (Extended Data Fig. 3)[42]. Further studies on the specific role of H4K16ac in neuronal cell types will reveal whether H4K16ac dysregulation could contribute to neuronal-specific dysregulation of TEs and gene expression, contributing to neurodevelopmental and neurodegenerative disorders.

In yeast, H4K16ac regulates lifespan and cellular senescence[70]. Senescent cells show enrichment of H4K16ac in promoter regions of expressed genes[71]. Analysis of publicly available H4K16ac ChIP–seq data showed a dramatic loss of H4K16ac across L1s and LTRs in senescent cells in comparison with proliferating cells (Extended Data Fig. 9), suggesting that proliferating cells, compared with replicative senescent cells, have adapted to the permissive chromatin state at TEs. However, by contrast, L1 elements are known to be transcriptionally derepressed during cellular senescence and to activate the interferon I (IFN-I) response[72]. Further investigation will be needed to understand the direct role of the H4K16ac pathway in regulating L1 transcription linked to aging and senescence.

In summary, we show that H4K16ac-marked L1s and LTRs act as enhancers to regulate genes in *cis*. The act of transcription at L1 5′ UTRs and LTRs mediated by H4K16ac could contribute to chromatin topology and enhancer-mediated regulation of host gene expression in *cis*, as L1 and LTRs that are marked with histone acetylations are located within the regulatory elements, or they interact with genes. The permissive chromatin structure mediated by H4K16ac and H3K122ac could counteract the epigenetic repressive environment at TEs within the regulatory elements (Fig. 6e)[73].

## Online content

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

## Methods

### Cell culture and transduction

The H9 hESC line was a gift from L. Vallier's lab with the MTA from WiCell. hESCs were grown on geltrex-coated plates (Thermo Fisher Scientific, A1413302) in mTeSR Plus medium (Stem Cell Technologies, 100-0276) supplemented with 100 U ml⁻¹ penicillin–streptomycin (Gibco, 15140122) and passaged every 3–4 d with ReLeSR (StemCell Technologies, 100-0484), according to the manufacturer's protocols. The doxycycline-inducible SpCas9 (iCas9-H1) hES cells were generated using parental H1-hESCs from WiCell. Briefly, H1 cells were transfected with plasmids from the Genome-CRISP Inducible Cas9 human AAVS1 Safe Harbor Knock-in Kit (GeneCopoeia) using Fugene HD (Promega) and selected with Puromycin (500 ng ml⁻¹). Cells were single-cell sorted using FACS and grown in mTESR to make monoclonal lines. The resulting SpCas9 line was confirmed to be karyotypically normal and was tested for mycoplasma every 3 weeks.

Transformed dermal fibroblasts (TDF) expressing guide RNAs (3 guides per pool) targeting MSL1 and MSL3 and parental (WT) TDF lines were generated in P. Scaffidi's lab (The Crick Institute). Cells were grown in MEM (Gibco, 11095080) supplemented with 10% FBS (Sigma, F7524), 1× Glutamax (Gibco, 35050061), 1× non-essential amino acid solution (Sigma, M7145) and 100 U ml⁻¹ penicillin–streptomycin.

iCAS9 cells were transduced with three lentiviral guide RNAs targeting MSL1 and MSL3 (ref. [54]). Parental iCAS9 H1, iCAS9 with MSL guide RNAs, TDF iCas9 transduced with MSL1, and MSL3 guide RNA pools were treated with 1 μg ml⁻¹ doxycycline (Sigma) to generate the inducible MSL-KO lines. After 4 to 7 d of doxycycline induction, the knockout was validated by immunofluorescence followed by high-content microscopy and western blot using antibodies to H4K16ac and H3K27ac.

HEK293T cells were grown in DMEM, high glucose (Lonza, BE12-614Q), supplemented with 10% FBS (Sigma, F7524), 1× Glutamax (Gibco, 35050061) and 100 U/ml penicillin–streptomycin. HEK293 and HeLa cells were grown in DMEM, high glucose (Lonza, BE12-614Q) supplemented with 10% FBS (Sigma, F7524), 1× Glutamax (Gibco, 35050061) and 100 U ml⁻¹ penicillin–streptomycin. PC3 and LNCaP cells were grown in RPMI medium (Gibco, 21875034) supplemented with 10% FBS (Sigma, F7524) and 100 U ml⁻¹ penicillin–streptomycin. RWPE1 cells were grown in a keratinocyte serum-free medium (Gibco, 10724011) supplemented with 100 U ml⁻¹ penicillin–streptomycin. K562 cells were grown in Iscove's Modified Dulbecco's Medium (Lonza, BE12-722F) supplemented with 10% FBS (Sigma, F7524) and 100 U ml⁻¹ penicillin–streptomycin. SH-SY5Y cells were grown in DMEM/F12 (1:1) medium (Gibco, 11320033) supplemented with 10% FBS (Sigma, F7524) and 100 U ml⁻¹ penicillin–streptomycin. All the cell lines were tested for mycoplasma contamination using EZ-PCR Kit (Geneflow, K1-0210).

For the generation of MSL3 stable knockdown H9 hESCs, cells were transduced with lentiviral particles (Sigma, Mission shRNAs, MSL3 sh1 TRCN0000022105, MSL3 sh2 TRCN0000022107) and mammalian nontargeting shRNA (SHC002V) at an MOI of 6. At 48 h after transduction, cells were selected with 0.5 μg ml⁻¹ puromycin (Gibco, A1113803) for 48 h, and surviving cells were then allowed to recover until they formed viable colonies.

### Western blotting

Cells were pelleted by centrifugation at 228g for 5 min at 4 °C and resuspended in RIPA buffer (150 mM sodium chloride, 1.0% NP-40, or Triton X-100, 0.5% sodium deoxycholate, 0.1% SDS (sodium dodecyl sulfate) and 50 mM Tris, pH 8.0) and protease inhibitors with benzonase (Novagen; final concentration, 1.25 U μl⁻¹) and incubated for 30 min on ice with intermittent mixing. Extracts were sonicated for 5 cycles with Bioruptor (Diagenode) with the 30 s on and 30 s off cycles, and were cleared by centrifugation at 15,500g for 10 min at 4 °C. Equal amounts of protein extract were denatured in 1× Bolt LDS sample buffer (Thermo Fisher Scientific, B0007) and separated on Bolt Bis-Tris gels

(Thermo Fisher Scientific, NW04120BOX, NW00122BOX), blotted on a polyvinylidene fluoride (PVDF) membrane (BioRad, 1704156) and immunoblotted with antibodies to MSL3 (Merck Millipore, ABE467, 1:1,000 dilution), L1 ORF1 (Merck Millipore, MABC1152, 1:1,000 dilution), H4K16ac (Abcam, ab109463, 1:5,000 dilution), H3K27ac (Abcam, ab4729, 1:5,000 dilution), α-tubulin (Sigma, T9026, 1:5,000 dilution), and HERV (Novus Biologicals, NB100-93579, 1:500 dilution), and horseradish peroxidase (HRP)-conjugated goat anti-rabbit IgG H&L (Abcam, ab6721) and HRP-conjugated goat anti-mouse H&L (Thermo Fisher Scientific, 31430) secondary antibodies.

### Immunofluorescence and imaging

Cells were grown on 24-well cell culture plates, fixed with 4% formaldehyde, incubated for 5 min with permeabilization buffer (PBS containing 0.1% Triton X-100), and blocked with PBS containing 0.1% Triton X-100 and 2% BSA) for 1 h. Primary antibodies to H4K16ac (Abcam, Ab109463, 1:500) and L1 ORF1 (Merck Millipore, MABC1152, 1:500 dilution) were added overnight at 4 °C, washed three times with PBS (10 min each) and incubated with anti-rabbit secondary antibodies (Abcam, Ab150080, 1:500) and DAPI (1:1000). After washing 3 times with PBS (10 min each), the cells were left in PBS and imaged with Incell2000.

### CUT&Tag

CUT&Tag was performed according to Kaya-Okur et al.[39] protocol with modifications to tissue processing, as described below. Experiments were performed in biological duplicates from each cell type. Approximately 100,000 cells were pelleted by centrifugation for 3 min at 600g at room temperature and resuspended in 500 μl of ice-cold NE1 buffer (20 mM HEPES-KOH pH 7.9, 10 mM KCl, 0.5 mM spermidine, 1% Triton X-100, and 20 % glycerol and cOmplete EDTA free protease inhibitor tablet) and were left to sit for 10 min on ice. Nuclei were pelleted by centrifugation for 4 min at 1,300g at 4 °C, resuspended in 500 μl of wash buffer, and held on ice until beads were ready. The required amount of BioMag Plus Concanavalin-A-conjugated magnetic beads (ConA beads, Polysciences) was transferred into the binding buffer (20 mM HEPES-KOH pH 7.9, 10 mM KCl, 1 mM CaCl₂ and 1 mM MnCl₂) and washed once in the same buffer; each time they were placed on a magnetic rack to allow the beads to separate from the buffer and resuspended in binding buffer. Then, 10 μl of beads was added to each tube containing cells and rotated on an end-to-end rotator for 10 min. After a quick spin to remove liquid from the cap, tubes were placed on a magnet stand to be cleared, the liquid was withdrawn, and 800 μl of antibody buffer containing 1 μg of the following primary antibodies was added: normal rabbit IgG (Santa Cruz Cat no sc-2027), H3K27ac (Abcam, ab4729), H4K16ac (Abcam, ab109463), H3K122ac (Abcam, ab33309), H3K4me1 (Abcam, ab8895), H3K36me3 (Abcam, ab9050)) H3K4me3 (Millipore, 07-473), H3K27me3 (Abcam, ab192985) and H3K9me3 (Abcam, ab176916)). The mixture was incubated at 4 °C overnight in a nutator. Secondary antibodies (guinea pig α-rabbit antibody, Antibodies online, ABIN101961) were added 1:100 in Dig-wash buffer (5% digitonin in wash buffer), and 100 μl was squirted in per sample while they were gently vortexed, to allow the solution to dislodge the beads from the sides, followed by incubation for 60 min on a nutator. Unbound antibodies were washed in 1 ml of Dig-wash buffer three times. Then, 100 μl of (1:250 diluted) protein-A-Tn5 loaded with adapters in Dig-300 buffer (20 mM HEPES pH 7.5, 300 mM NaCl, 0.5 mM spermidine with Roche cOmplete EDTA free protease inhibitor) was added to the samples, placed on nutator for 1 h and washed three times in 1 ml of Dig-300 buffer to remove unbound pA-Tn5. Next, 300 μL Tagmentation buffer (Dig-300 buffer + 5 mM MgCl₂) was added while being gently vortexed, and samples were incubated at 37 °C for 1 h on an incubator. Tagmentation was stopped by adding 10 μl 0.5 M EDTA, 3 μl 10% SDS, and 2.5 μl 20 mg ml⁻¹ Proteinase K to each sample. Samples were mixed by full-speed vortexing for ~2 s and incubated for 1 h at 55 °C to digest proteins. DNA was purified by phenol:chloroform extraction using phase-lock tubes (Quanta Bio)

followed by ethanol precipitation. Libraries were prepared using NEB-Next HiFi 2× PCR Master mix (M0541S) with a 72 °C gap-filling step, followed by 13 cycles of PCR with 10-second combined annealing and extension for the enrichment of short DNA fragments. Libraries were sequenced in Novaseq 6000 (Novogene) with 150 bp paired-end reads.

## RT–qPCR
Total RNA was isolated from H9 hESCs using TRIzol reagent (ThermoFisher Scientific, 15596026). For RT–qPCR, cDNAs were prepared with LunaScript RT SuperMix Kit (NEB, E3010). For CRISPRi experiments, RNA isolation was done using a kit (Monarch, T2040S) followed by reverse transcription using LunaScript RT SuperMix Kit (NEB, E3010), qPCR using qPCRBIO SyGreen Mix Lo-ROX (PCRBio) in LightCycler 480 instrument (Roche). The list of specific primers used is given in Supplementary Table 4. RT–qPCR was done with three independent biological replicates, each of control shRNA and two independent shRNAs targeting MSL3 or relevant empty vector controls and dCAS9 systems for CRISPRi, on a StepOnePlus Real-Time PCR System (Applied Biosystems). Data were normalized to β-actin from three biological replicates.

## RNA sequencing
RNA was isolated using Monarch RNA mini prep kit (NEB) with genomic DNA elimination column and on-column DNase treatment. MSL3 KD RNA sequencing libraries were prepared by spiking in equal amounts of The External RNA Controls Consortium (ERCC) Spike-in RNA Variant Control Sets (SIRV set 3, Lexogen), and 500 ng of RNA was used for depletion of rRNA using RiboCOP kit (Lexogen), followed by RNA-seq library preparation using CORALL Total RNA-Seq Library Prep Kit (Lexogen). Libraries were sequenced as 150 bp paired-end reads using Novaseq 6000. In the case of H1 iCAS9 and MSL1 KO RNA-seq, Ribosomal RNAs were depleted using NEBNext rRNA Depletion Kit (Human/Mouse/Rat) (NEB no. E7400) followed by library preparation using NEBNext Ultra II Directional RNA Library Prep Kit for Illumina (NEB no. E7765).

## ATAC-seq
ATAC-seq was performed as described in ref. 23, with modifications. The freshly collected 50,000 cells were washed in PBS and resuspended in a resuspension buffer (10 mM Tris-HCl, 10 mM NaCl, 3 mM MgCl2). Cells were resuspended and incubated on ice for 3 min in 50 µl of cold lysis buffer (0.1% NP-40, 0.1% Tween-20, 0.01% digitonin in resuspension buffer). Nuclei were washed in 1 ml of wash buffer (990 µl resuspension buffer, 0.1% Tween-20) by inversion three times. Nuclei were pelleted by centrifugation at 500$g$ for 10 min at 4 °C. The nuclei were resuspended in 47.5 ml of Nextera Tagmentation buffer (Nextera DNA Sample Preparation Kit) and incubated with 2.5 µl of the Tn5 transposase (Nextera kit, Illumina) at 37 °C for 30 min. The resulting DNA fragments were purified using a miniElute column (Qiagen) and amplified by NEBNext High-Fidelity PCR Master Mix in a total volume of 50 µl. The thermocycling protocol for this reaction was 72 °C for 5 min, 98 °C for 30 s and five cycles of 98 °C for 10 s, 63 °C for 30 s, and 72 °C for 1 min. The universal adapter primer and a unique barcoded adapter primer (same as CUT&Tag primers) were used. To avoid over-amplification, after the initial five cycles, the number of remaining cycles required was estimated for each sample using qPCR by adding SYBRGreen and using 5 µl of the previous PCR as a template. The number of additional cycles was determined to be the number that it took for the qPCR to reach one-third of maximal fluorescence. The original PCR was then resumed, and each sample was cycled as necessary. After amplification, the samples were purified using AMPure XP beads. The libraries were sequenced as a minimum of 50 million 150 bp paired-end reads in Novoseq (Novogene PLC).

## CRISPRi with dCAS9-KRAB
CRISPRi using dCAS9-KRAB was performed as described in ref. 31, with the following modifications. The CRISPR-Bac plasmid

(PB_tre_dCas9_KRAB, Addgene ID 126030) (ref. 74), a kind gift from J. Mauro Calabrese, was mixed with the piggyBac-transposase plasmid in a 1:1 ratio (2 µg in each well of a 6-well plate) into opti-MEM, along with TransIT-LT1 in a 1:3 ratio (Mirus, MIR2300), and reverse transfected into H9 hESCs according to manufacturer's protocol. The next day, the cells were allowed to recover from the transfection for 24 h and then selected with 100 µg ml$^{-1}$ hygromycin B for 5 d. Surviving colonies were then expanded and reverse-transfected with various gRNA-expressing plasmids (cloned into pSLQ1371 as described in ref. 75, kind gift from S. Qi) with TransIT-LT1. Then, 1.25 × 10$^6$ cells were reverse transfected with 1 µg of the gRNA-expressing plasmid (per well of a 24-well plate). To improve the efficiency of plasmid delivery, the transfection was repeated the next day (forward transfection). At 48 h after the first transfection, cells were briefly selected with puromycin (0.5 µg ml$^{-1}$) for 24 h and left to recover for 96 h. Cells were collected for RNA isolation and RT–qPCR.

## CRISPR–Cas9 deletion of LINE1 elements in hESCs
Two crRNAs performed LINE1 element deletions and were designed to target nonrepetitive flanking sites of the LINE1 elements (Supplementary Table 5). Individual crRNAs were mixed with tracerRNAs Alt-R CRISPR–Cas9 tracrRNA, ATTO 550, and with CAS9 protein (Alt-R S.p. HiFi Cas9 Nuclease V3) to form ribo-nucleocomplex. Then, 200,000 H1 hESCs per well were nucleofected in the presence of Alt-R Cas9 Electroporation Enhancer in 16 strips format using primary cell kit (P3). hESCs were electroporated using a 4D nucleofector, the P3 Primary Cells 4D-Nucleofector X kit S (Lonza, LOV4XP3032), with the pulse program. After nucleofection, cells were resuspended in an hESC medium supplemented with ROCK inhibitors and seeded to geltrex-coated 96 wells for 2 d at 37 °C in a humidified incubator with 5% CO$_2$. hESCs were split into 96 wells and 6-well plates for picking of single-cell colonies. The pool of cells 5 d after nucleofection was collected to check the deletion efficiency and for RT–qPCR. Cells were seeded in 6-well plates for picking single-cell colonies; the deletion was assessed by rapid DNA lysis and PCR using PCRBIO Rapid Extract PCR kit (PB10.24-40). For deletion of L1s at MOXD1 and RLN2 locus, pools of cells that were collected 5 d after nucleofection were used for RT–QPCR. PCR products were subjected to Sanger sequencing. Primer sequences used for screening are listed in Supplementary Table 4.

## Analysis of CUT&Tag-seq data
**Mapping.** For the CUT&Tag-seq, 150-bp paired-end reads were trimmed for adapters using the Trimmomatic tool and aligned to the hg38 genome through local Bowtie2 (version 2.4.5) with these parameters for pair-end mapping:–very-sensitive-local–no-unal–no-mixed–no-discordant–phred33 -I10 -X 700 (ref. 76). For analyzes, multi-mapped reads were filtered out, and only uniquely mapped reads were retained with the samtools flag of -q 2 -f 0x200 (ref. 77). For Figure 5e, total reads, including multi-mapped reads, were retained for plotting heatmaps of ATAC-seq, CUT&Tag, and RNA-seq reads. For individual replicates, the bam files were sorted, indexed, and used for generating bedgraphs (for peak calling) and bigwigs. The bam files were sorted and indexed using the samtools (version 1.9) sort and the samtools index. Merging of multiple replicates was performed using samtools merge. The sorted bam files were used to generate bed, bedgraph and bigwig formats for individual modifications.

**Peak calling and analyzes.** The reads were extracted from the bam to bed by the bedtools bamtobed option[78]. Further reads were processed as mentioned in the SEACR (version 1.3) manual to get the bedgraph[79]. These bedgraph files were subjected to peak calling through SEACR with a stringent $P$ of ≤1 × 10$^{-6}$ with the norm and relaxed options.

Further bedtools with various options were used for transforming bed files, such as intersect, closest, sample, or shuffle. GNU awk editor was used for processing the bed files wherever required. Chromatin

state predictions for the histone modification peaks were performed using ChromHMM[40] (v1.10). For further analyses, the reproducible peaks were obtained by performing an intersection between peak files for each CUT&Tag replicates for histone PTMs. Overlap between the peaks for histone modifications for the H9 cells was assessed using the Intervene package[80]. While the overlapping peak counts were plotted as a Venn Diagram for each histone PTM and IgG, the peaks for histone PTM combinations were plotted as an upset plot showing the number of overlapping peaks (y axis) along with the histone modification peak numbers on the x axis.

**TE enrichment analyses.** Tracks for the repeats (rmsk) were obtained from the UCSC Genome Table Browser for hg19 and hg38. Reproducible peaks (consistent between two replicates) were used to generate the observed versus expected frequency for different TE classes (Alu, full-length L1, and LTR), gene body, and TSSs. These were calculated across various histone modification CUT&Tag peaks. The intersect count was obtained for each histone modification using bedtools (version v2.28.0) intersect (bedtools intersect -wa -u options) for the mentioned genomic elements. The expected occurrences in the genome were calculated by intersecting the genomic elements with the randomized genomic coordinates (number, length, and chromosome ID matched) across different histone modifications. The ratios for observed versus expected at these genomic elements for each histone PTM were calculated and used to plot as a heatmap using ggplot2 heatmap function in R.

**RepeatMasker.** To analyze the repeat content of the different histone modification peak sets, the fasta was obtained using the bedtools getfasta tool from the hg38.fa reference genome. The sequences were subjected to RepeatMasker (version 4.0.7) to get the repeat content across these genomic sequences for different histone modifications (http://www.repeatmasker.org).

**Bigwig generation and plotting.** Sorted bam files were subjected to bigwig generation via deepTools (version 3.5.1) (ref. [81]) bamCoverage tool with –binSize 20 –normalizeUsing BPM or CPM –scaleFactor = 1.0 –smoothLength 60 –extendReads 150 –centerReads options. The signal was normalized to IgG through bigwigCompare with option –operation first or subtract. The bigwig files were used for plotting signals or visualization in the genome browser. The genome-browser views were obtained by viewing the signal tracks in the UCSC Genome Browser. For the knockdown (in H9 cells) or knockout (in TDF cells) studies, the samples were normalized on the basis of the number of reads mapping to the *Escherichia coli* genome.

The plotting of signals at various genomic landmarks and bed coordinates was carried out through deepTools. Matrices were generated using deepTools computeMatrix reference-point or scale-regions option. These matrices were used for plotting heatmaps or average summary plots by the plotHeatmap or plotProfile function in deepTools, with or without clustering by the k-means algorithm. The sorted bam files were also used to study the correlation between the individual replicates for the CUT&Tag across histone PTMs and IgG using multiBamSummary function in deeptools with options bins and plotted as Pearson correlation heatmap using deeptools plotCorrelation function with options –skipZeros.

Further, H4K16ac signals from GSE84618 for brain (prefrontal lobe) tissues from young individuals, old individuals and individuals with Alzheimer's disease were compared for the TE elements. Similarly, the bigwigs were obtained for the proliferative and senescence model in IMR90 cells (GSM1358821) for L1 and LTR subfamilies. The signals were compared as heatmaps or average summary profiles using above-mentioned tools. The signals at the H4K16ac-marked TEs were also plotted as average summary plots using plotProfile function for transcription factors YY1 (ENCODE

ID: ENCFF904SDR), RAD21 (ENCODE ID: ENCFF506AAX) and CTCF (ENCODE ID: ENCFF473IZV) using ENCODE datasets (bigwigs) normalized as fold change over control.

**TAD border annotation.** To call TADs in human embryonic stem cells (H9), we used Hi-C data for two replicates from ref. [53] under accession numbers (GSM3262956 and GSM3262957). We first generated contact domains for all chromosomes at a 10-kb resolution using the Arrowhead tool from Juicer using Knight-Ruiz Normalization[82]. We extracted the borders of these TADs. To ensure we identify a robust set of TAD borders, we selected with a score above one that are common borders between the two replicates, assuming a maximum gap of 1 bin (10 kb). This resulted in 9,952 robust TAD borders. The TAD calling was performed on the hg19 reference genome, and to allow integration with the rest of the analysis, we lifted over the common TAD borders from hg19 to hg38.

**Significant loops calling using Micro-C.** Chromatin loops were called with the HiCCUPS tool from the Juicer software suite[82] on micro-C data in H1 hESCs[51]. Loops were called using 5- and 10-kb resolution, 10% FDR, Knight-Ruiz normalization, a window of 7 and 5, peak width of 2 and 4 and, thresholds for merging loops of 0.02, 1.5, 1.75 and 2, and distance to merge peaks of 20 kb (–r 5000,10000 -k KR -f .1,.1 -p 4,2 -i 7,5 -t 0.02,1.5,1.75,2 -d 20000,20000).

**Motif enrichment analysis.** Enrichment of TF-binding sites (TFBS) at the TEs (>5 kb L1, Alu, and LTR) overlapping with histone modifications (H4K16ac, H3K27ac and H3K122ac) or a similar number of randomized genomic bins (chromosome, length matched) was performed. The experimentally determined TFBS for the H1-hESCs was fetched from the UCSC Genome Browser as TFBS clusters. The number of motifs for each TE class was either positive for histone modification or randomized genomic bins for histone modification for all TFBS. The internal distribution profile of motifs across each TEs was determined as percentage distribution and enrichment score defined as (Diff/Sum) of motif counts' percentage between observed (TEs positive for histone modification) versus expected (randomized genomic bins) occurrence of motifs. The ratios obtained for each TFBS were plotted as a heatmap using the R package ComplexHeatmap[83].

**ATAC-seq data analysis.** The ATAC-seq reads were processed for mapping by trimming for adapters using the Trimmomatic tool, followed by aligning to the hg38 genome through local Bowtie2 (version) with these parameters for pair-end mapping: –very-sensitive-local –no-unal –no-mixed –no-discordant –phred33 -I 10 -X 700 (Alteration in -X to 2000 was done to allow the mapping of reads for H9 cells). The mapped reads were processed as described above (CUT&Tag data analysis) to generate the bigwigs. The signal was normalized as $\log_2$(fold change) for control over MSL3 knockdown using the bigwigCompare function in deeptools with –skipZeroOverZero –operation log2. Using the same matrix generation and heatmap tools, further ATAC-seq signals were compared at the full-length L1 subfamilies and LTR subclasses.

**RNA-seq data analysis.** The reads obtained from RNA-seq for H9 cells were mapped to the human genome using STAR[84] following the Bluebee-CORALL pipeline of mapping. For TDFs, the RNA-seq datasets were downloaded from NCBI-GEO for accession ID GSE144019. The reads were mapped to hg38 following the same pipeline as H9 except for the single-end specification in TDFs.

For differential enrichment analysis, the fragment counts for each dataset were obtained using the featurecounts tool from the SubRead package. The GTF file for genes was obtained from ENSEMBL, and for different TE classes (Alu, L1 and LTR), it was fetched from the UCSC

Table Browser. These feature counts were used for the differential enrichment analyses using the DESeq2 package in R. The DESeq was performed with defaults[85]. The differential expression of genes was visualized as a volcano plot. The differential gene expression table can be found in the additional data.

The uniquely mapped reads were filtered using samtools for MAPQ of 255. Further, the unique alignments' bam files were merged, sorted, and indexed using samtools, followed by bigwig generation using the deepTools function bamCoverage. The normalized signal was generated as $\log_2$(fold change) for control over MSL3 knockdown using bigwigCompare function in deeptools with −operation log2. The signal was compared at the full-length L1, as well as at genes.

The RNA signals across the various subfamilies of TEs, as well as genes in the flanks (<10,000, 10,000–25,000 and 25,000–50,000 kilo base (kb) of the H4K16ac-marked LTR and full-length L1, were calculated as RPKM from the read counts obtained for each gene or TEs across the multiple replicates for H9 (control or MSL3 KD) and TDF (WT or MSL1 KO). The RPKM signal was then plotted as a violin and box plot using ggplot2 in R. For comparison, the same number of TEs that are H4K16ac$^+$ and H4K16ac$^-$ in TDFs was obtained by subsampling using the bedtools sample. The signal was plotted as the log10 value of the RPKM on the $y$ axis. The RNA signals were plotted as violin plots with box plots with a median. The statistical analyses for all the violin plot comparisons were performed using the Dunn test with Bonferroni correction.

**STARR-seq data analysis.** To assess the potential of the TEs marked by H4K16ac to act as enhancers, we compared the STARR-seq signal in K562 (ENCFF611ZHY) and SH-SY5Y (ENCFF571ARG) cells at the TE elements: full-length L1 (>5 kb) and ERV/LTRs. The signal was plotted as a heatmap from the start (L1) or center (LTR) of the TE elements sorted according to the H4K16ac signal. Further, the signal was compared as violin plots for the four sets of peak combinations with respect to overlap among peaks that overlap with TEs. These were H3K4me1$^+$ only, H3K4me1$^+$H3K27ac$^+$H4K16ac$^+$, H4K16ac$^+$H3K4me1$^-$ and H4K16ac$^+$H3K4me1$^+$ peaks that overlap with LTRs.

**Statistical tests.** For all the RT–qPCRs, an unpaired $t$-test with Welch correction (two-stage step-up) was performed between the groups using GraphPad Prism9. The statistical tests were performed for all the violin plots using the Dunn test function in the R tool rstatix. Dunn's test with Bonferroni correction was used for multiple-group comparisons between the groups.

**Reporting summary**
Further information on research design is available in the Nature Portfolio Reporting Summary linked to this article.

## Data availability
The data discussed in this publication have been deposited in NCBI's Gene Expression Omnibus and are accessible through GEO Series accession number GSE200770 (https://www.ncbi.nlm.nih.gov/geo/query/acc.cgi?acc=GSE200770). CUT&Tag raw data as well as processed data files (peaks and bigwigs), can be accessed at NCBI under accession ID GSE200768, RNA-seq raw data files can be accessed under accession ID GSE200769, and ATAC-seq datasets can be accessed under accession ID GSE200767. All the datasets generated and public datasets used in this study are detailed in Supplementary Table 1. Source data are provided with this paper.

## Code availability
All the analyses in this manuscript have been carried out using publicly available tools. No custom code was generated for this purpose. The methodology contains the details of the analysis steps involved.

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

## Acknowledgements
We thank QMUL epigenetics hub members A. de Mendoza, C. Bell, V. Rakyan, and L. Stojic (QMUL) for discussing and reading the manuscript. We thank L. Vallier (Cambridge UK, with MTA from WiCell) for sharing the H9 cell line. We thank S. Henikoff (Fred Hutchinson Cancer Research Center), E. Schulz (Max Planck Institute for Molecular Genetics), and C. Martin (Queen Mary University of London) for sharing reagents. We thank P. Dubey, I. Alic, and A. Murrey (QMUL) for help with hESC cell culture. We thank G. Warnes and L. Gammon (Blizard Institute, QMUL) core facilities for help with the flow sorter and high-content imaging analysis. This research used Apocrita HPC, supported by QMUL Research-IT. Funding: Medical Research Council UKRI/MRC grant (MR/T000783/1) (M.M.P., D.P., M.P., F.B.), a Barts charity Rising Stars award and a Barts charity small grant (MGU0475) (M.M.P.), a Marie Skłodowska-Curie grant 896079 (J.S.), BBSRC (BB/T000031/1) (M.R.B.), and Cancer Research UK, UKRI/MRC, and the Wellcome Trust Welcome Trust (FC001152) (P.S.).

For open access, the author has applied a CC BY public copyright license to any author accepted manuscript version arising from this submission.

## Author contributions
M.M.P. acquired the funding, conceived and designed the study, and supervised the work. D.P. performed MSL3 knockdowns, M.S.L. iCAS9 knockout, RT–qPCR, CUT&Tag, CRISPRi, CRISPR deletions, RNA-seq, western blot, and imaging, with contributions from M.M.P., J.S, F.B and S.B. M.P. analyzed CUT&Tag, ChIP–seq, STARR-seq, and RNA-seq data, with contributions from D.P. and M.R.B. O.A.G. analyzed the Hi-C and Micro-C data with supervision from N.R.Z. and M.M.P. M.M.P. interpreted the data with contributions from M.P., D.P., N.R.Z., and M.R.B. S.D.M.S. and P.S. shared iCAS9 cell lines. M.M.P. wrote the

manuscript with contributions from D.P. and M.P. All the authors have approved the final version of the manuscript.

## Competing interests

The authors declare no competing interests.

## Additional information

**Extended data** is available for this paper at https://doi.org/10.1038/s41594-023-01016-5.

**Correspondence and requests for materials** should be addressed to Madapura M. Pradeepa.

**Peer review information** : *Nature Structural & Molecular Biology* thanks Ruchi Shukla and the other, anonymous, reviewer(s) for their contribution to the peer review of this work. Primary Handling Editor: Carolina Perdigoto and Dimitrios Typas, in collaboration with the *Nature Structural & Molecular Biology* team. Peer reviewer reports are available.

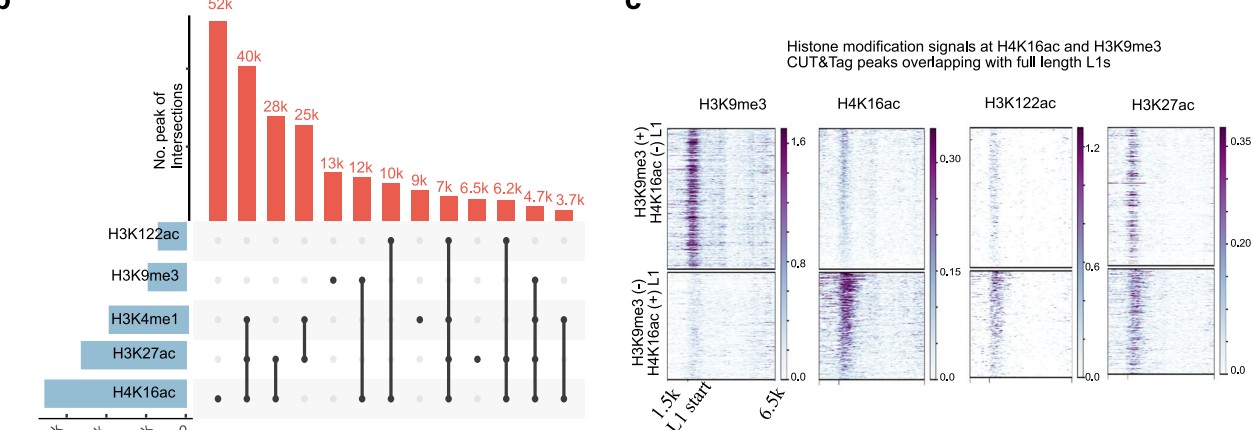

**Extended Data Fig. 1 | Related to Fig. 1. CUT&Tag data correlation data and overlap of histone modification peaks and at LINE1. a**. Pearson correlation heatmap for the CUT&Tag replicates across histone modifications in H9 cells. **b**. Upset plot showing the intersection of CUT&Tag peaks at TE (LTR, *Alu* and full-length L1) families. The X-axis shows the total number of peaks, and the Y-axis is the number of peaks intersected. **c**. Heatmaps showing signals (CPM) for the H3K9me3, H4K16ac, H3K27ac and H3K122ac at the full-length L1s marked by either H3K9me3 (top of each heatmap) or H4K16ac (bottom of each heatmap).

Intersection of CUT&Tag peaks from biological replicates for histone modifications in hESCs

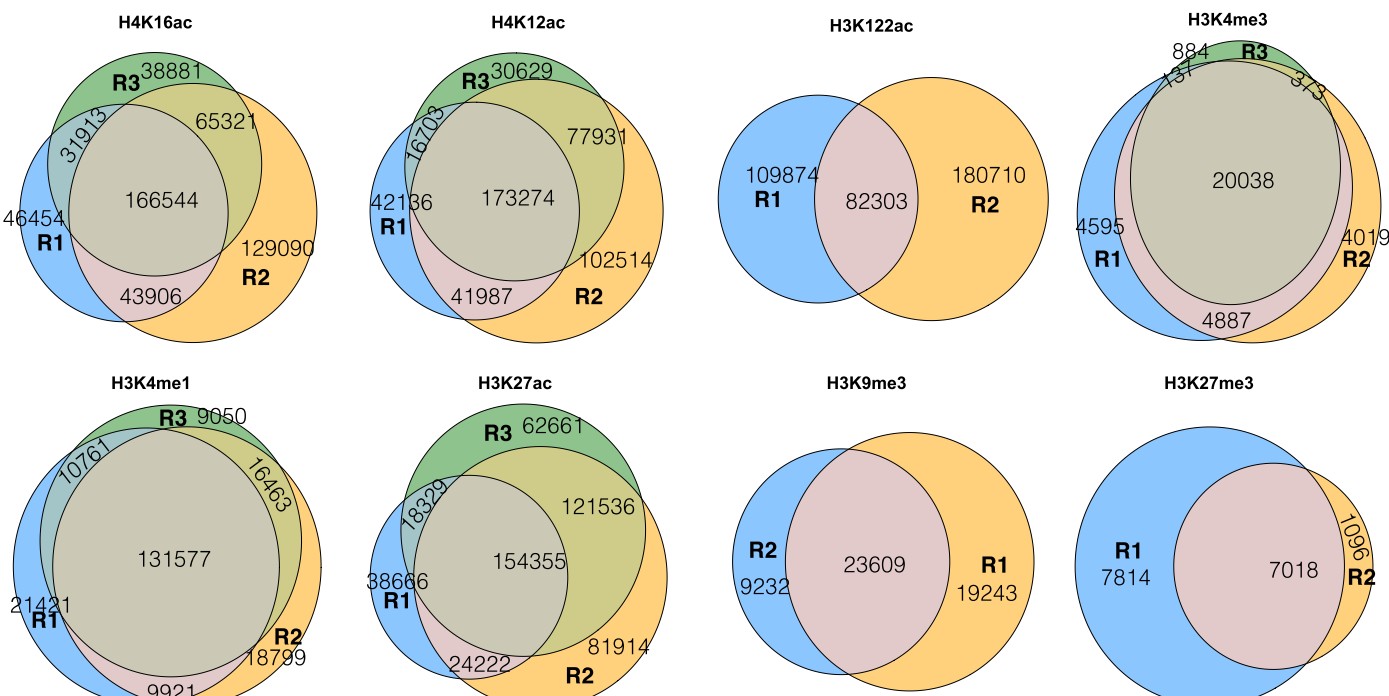

**Extended Data Fig. 2 | Related to Fig. 1, Overlap of CUT&Tag data peaks from replicates.** Venn diagrams showing reproducibility for the CUT&Tag peaks among the replicates called for the Histone PTMs.

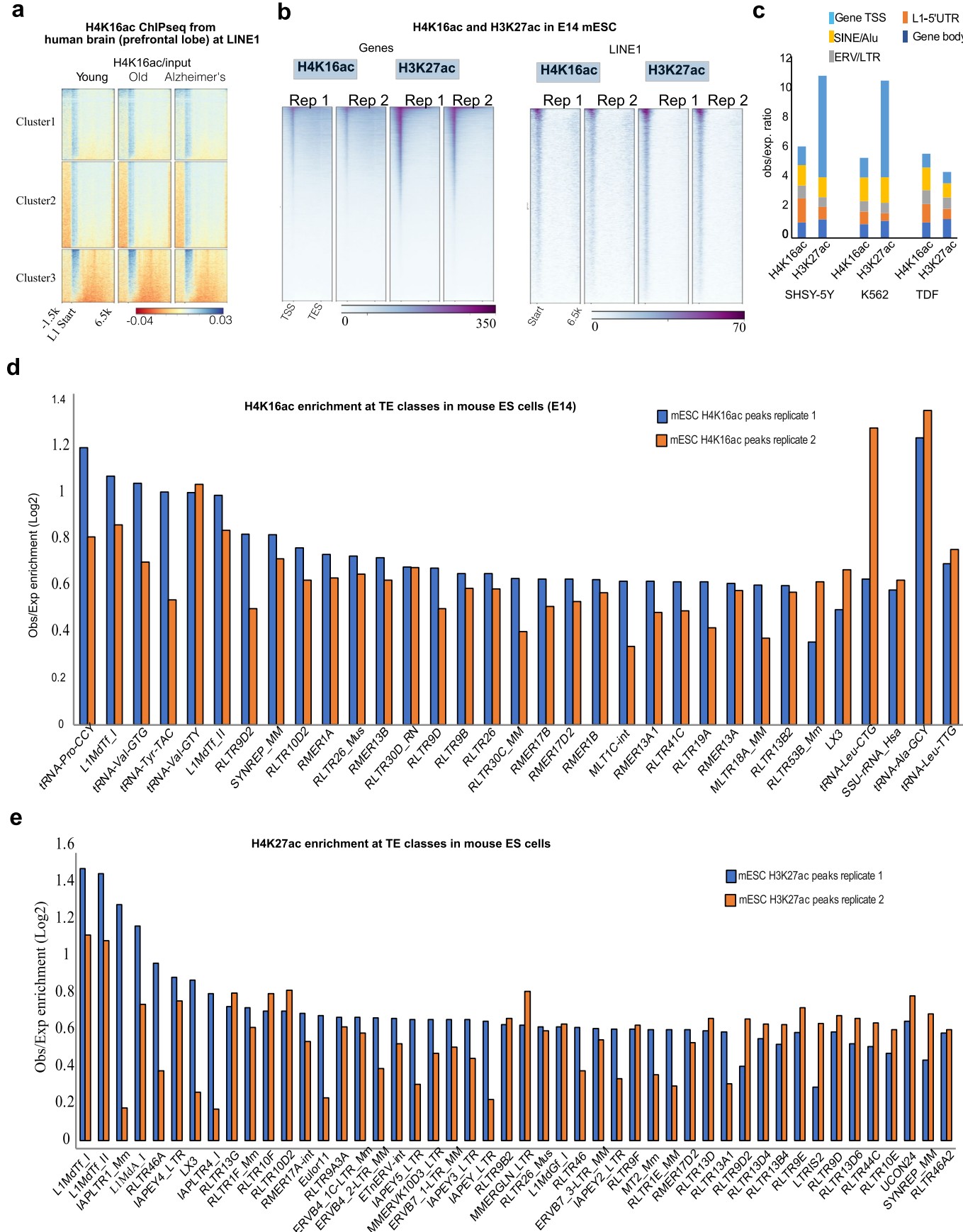

**Extended Data Fig. 3 | See next page for caption.**

**Extended Data Fig. 3 | Related to Figs. 1 and 2. H4K16ac is enriched at TEs in human brain, cancer and mouse stem cells. a**. Heatmap showing H4K16ac ChIPseq signal across full-length L1s in the human brain (prefrontal lobe) tissues from young, old and Alzheimer's patients from Nativio et al. 2018. **b**. Heatmap showing two replicates of H4K16ac and H3K27ac CUT&Tag signal across RefSeq genes (left) and full-length L1s (>5 kb) from the mouse genome. **c**. Stacked bar plot showing ratio (Y-axis) of observed over expected (background) for the TSS, gene body and TEs (LTR, Alu and L1) overlapping with H4K16ac or H3K27ac (X-axis) in SHSY-5Y, K562 and TDF cells. **d** and **e**. Observed over expected enrichment ratio for H4K16ac and H3K27ac mouse embryonic stem cells (E14 mESCs) CUT&Tag peaks at transposable elements from mouse genome (from Repbase).

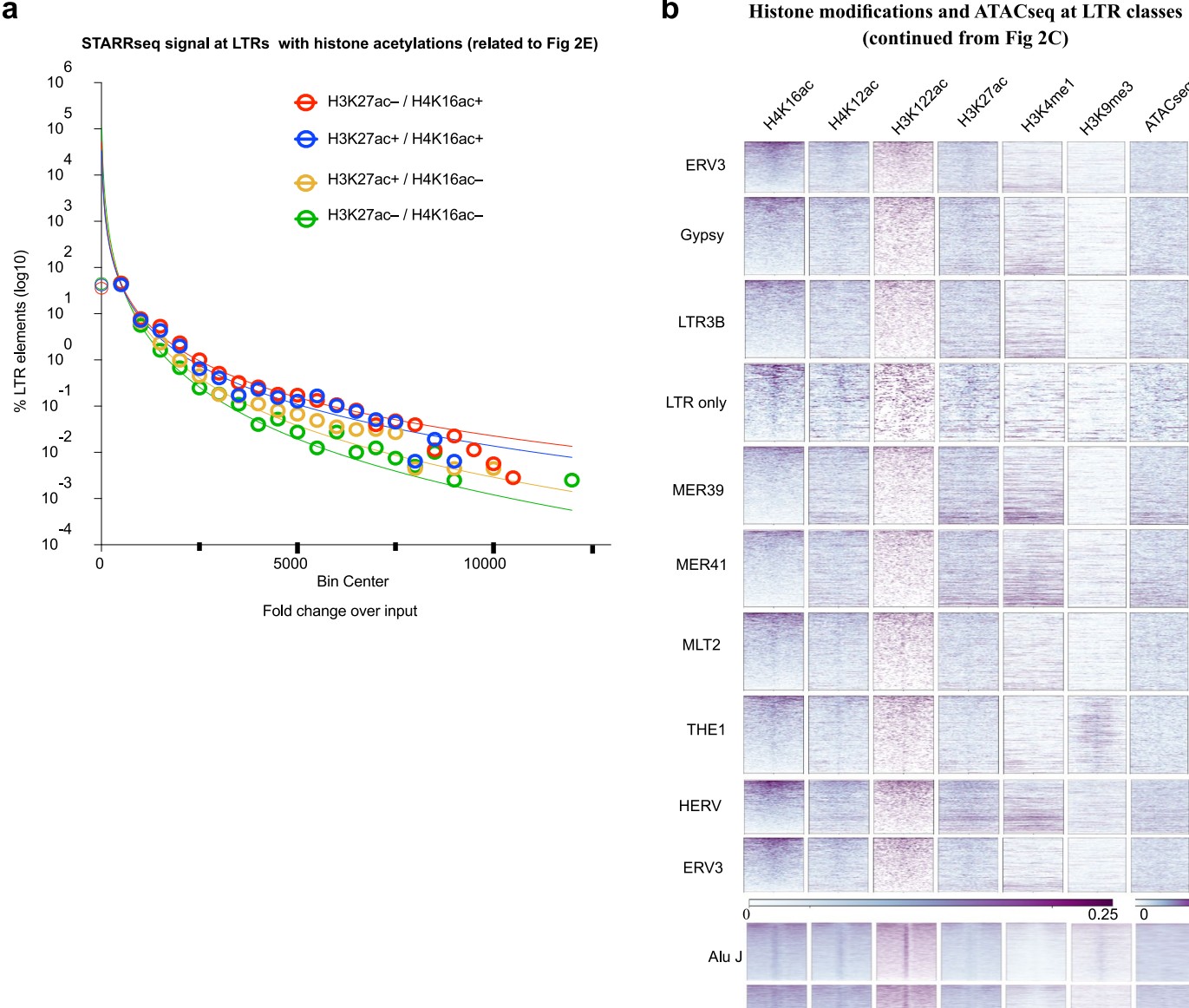

**a**

STARRseq signal at LTRs  with histone acetylations (related to Fig 2E)

**b**

Histone modifications and ATACseq at LTR classes
(continued from Fig 2C)

**Extended Data Fig. 4 | Related to Fig. 2. a STARRseq and histone mark.** Related to Fig. 2. **a**. Frequency distribution of LTR elements (Y-axis, log 10 percentage) showing the STARR-seq signal enrichment (X-axis) that are H3K27ac−/H4K16ac+, H3K227ac+/H4K16ac+, H3K227ac+/H4K16ac− or H3K227ac−/H4K16ac−.

**b**. Heatmaps showing signals (CPM) for the H4K16ac, H4K12ac, H3K27ac and H3K122ac, H3K4me1, H3K9me3 and ATACseq at the LTR subfamilies and Alu subfamilies.

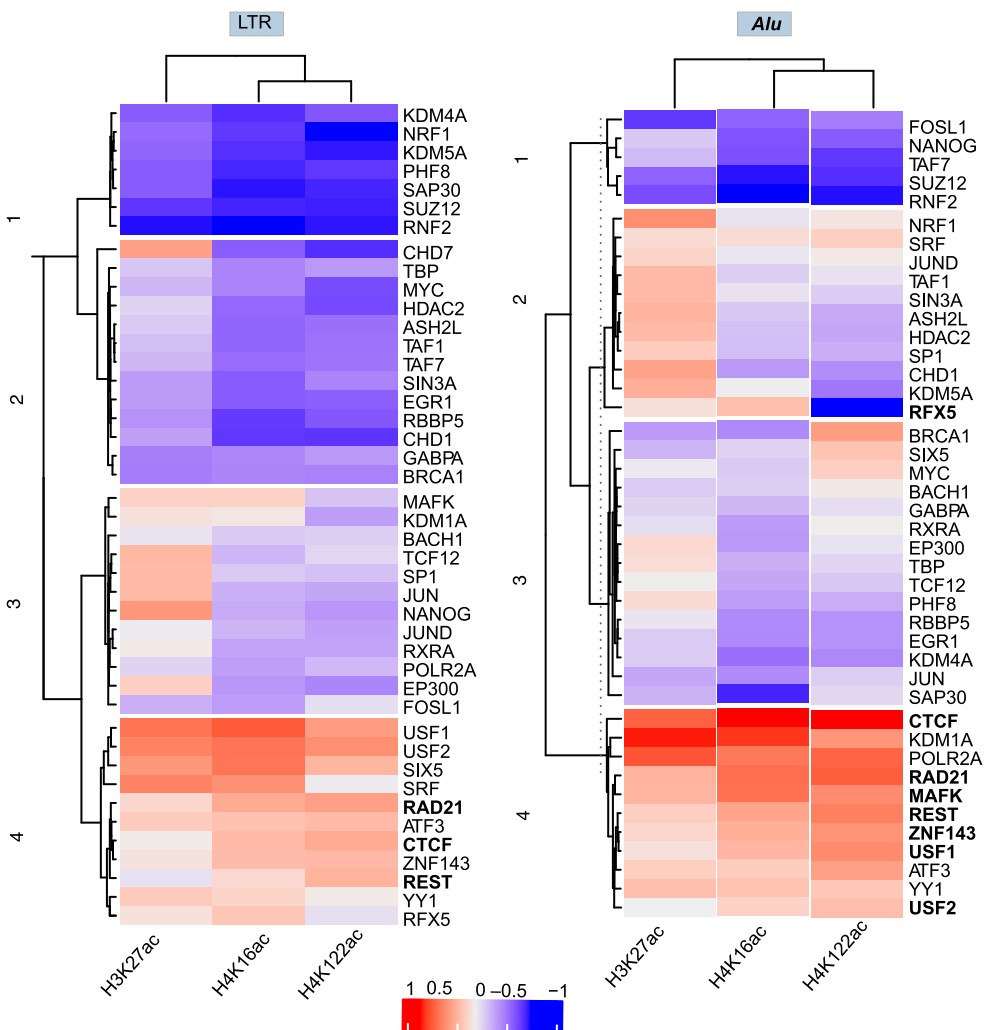

**Extended Data Fig. 5 | Related to Fig. 3. Continuation of Fig. 3.** Like Fig. 3a, transcription factor binding sites enriched at the H3K27ac, H4K16ac and H3K122ac marked LTR and *Alu* in hESCs.

**a**

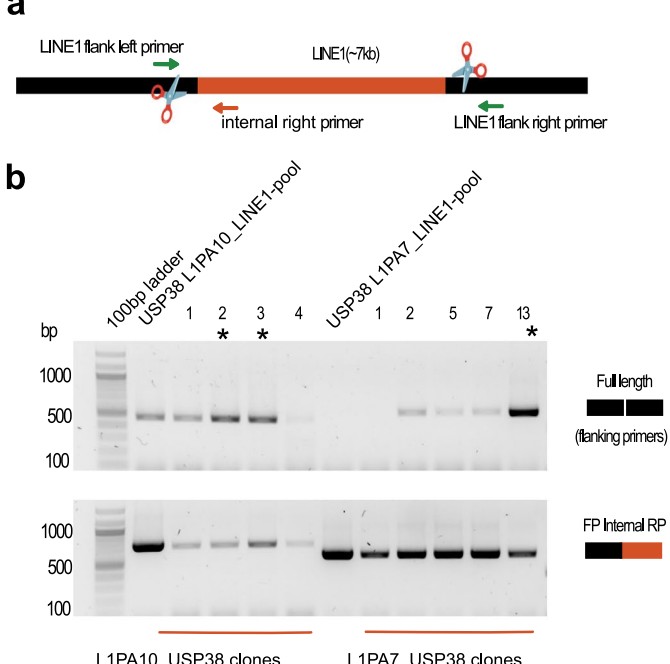

**b**

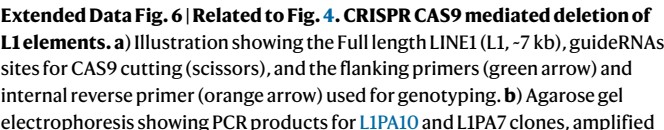

**c**

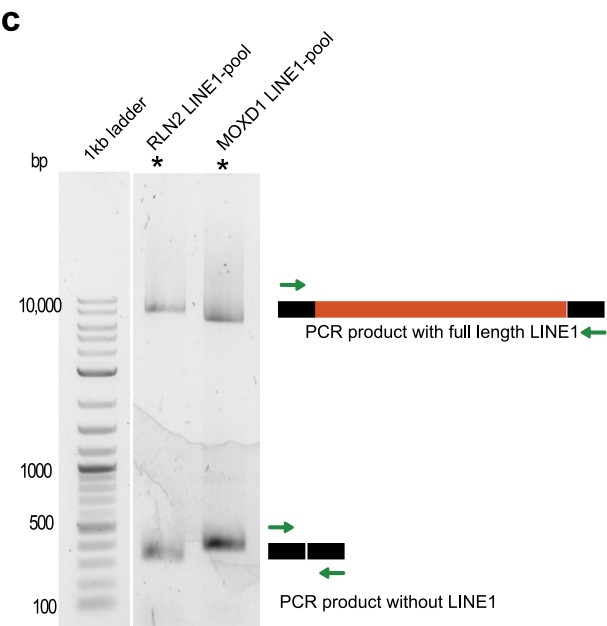

**Extended Data Fig. 6 | Related to Fig. 4. CRISPR CAS9 mediated deletion of L1 elements. a**) Illustration showing the Full length LINE1 (L1, ~7 kb), guideRNAs sites for CAS9 cutting (scissors), and the flanking primers (green arrow) and internal reverse primer (orange arrow) used for genotyping. **b**) Agarose gel electrophoresis showing PCR products for L1PA10 and L1PA7 clones, amplified using L1 flanking primers. ~500 bp amplification showing deletion of L1 (above). PCR with internal reverse primers showing presence of wild type allele (below). **c**) PCR amplicons with L1 flanking primers for pool of cells showing nearly 50% deletion efficiency for L1PA7 at the RLN2 locus and L1PA8 at the MOXD1 locus.

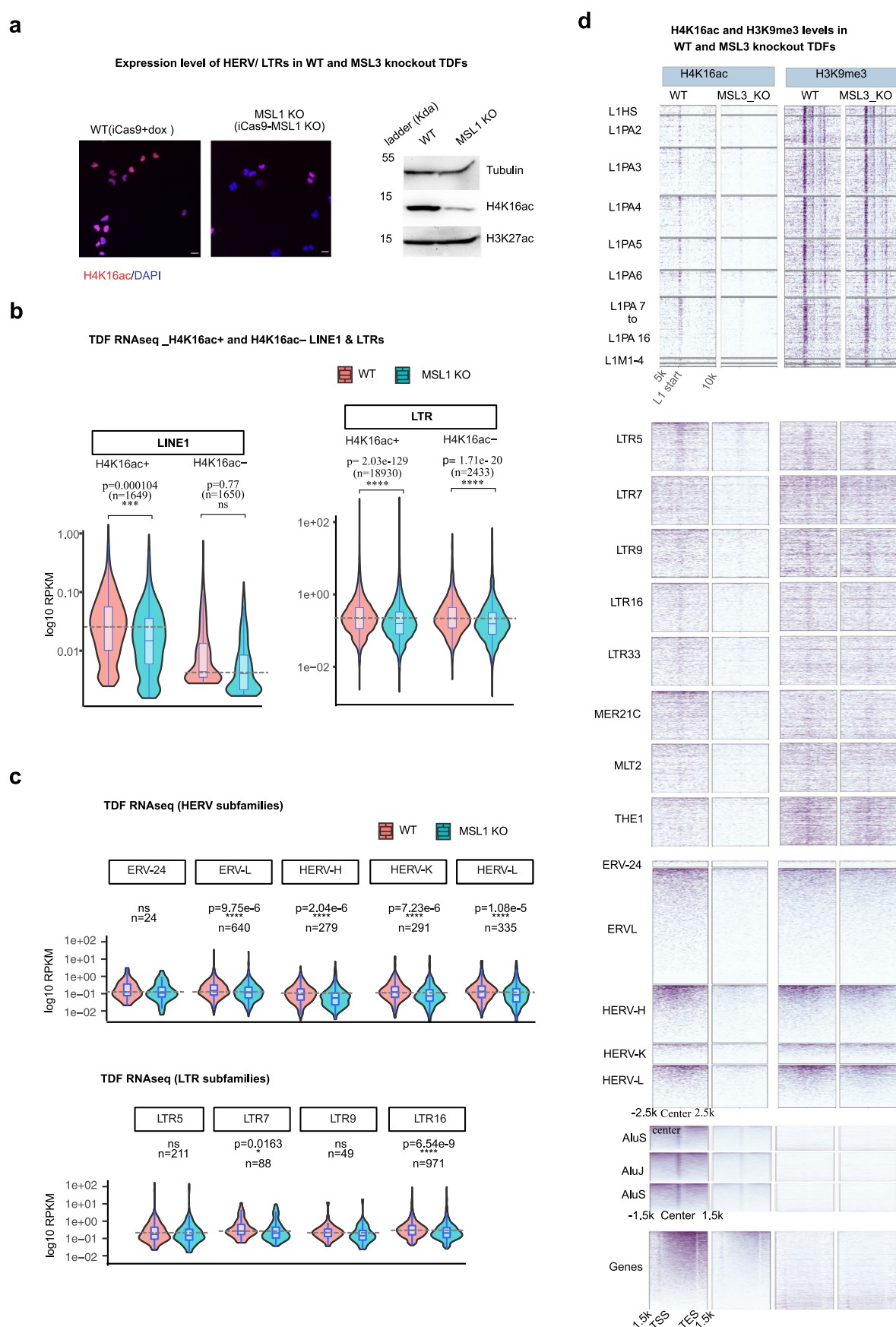

**Extended Data Fig. 7 | See next page for caption.**

**Extended Data Fig. 7 | Related to Fig. 5. Depletion of MSL proteins leads to downregulation of TEs. a**. Immunofluorescence images showing H4K16ac levels (Magenta) in WT and MSL1 KO TDFs (left). Western blots showing H4K16ac level in MSL1 KO and WT TDFs (Right). **b**. Violin plots showing the log10 RPKM signal of RNAseq reads for parental (WT) and doxycycline-inducible MSL1 (MSL1 KO) for L1s (left panel) and LTRs (right panel) that are either H4K16ac+ or H4K16ac. **c**. Violin plots for RNAseq signal across different ERV subfamilies (ERV24, ERVL, HERVK, HERVH and HERVL; top), and LTR subfamilies (LTR5, LTR7, LTR9 and LTR16; below). Statistical tests for all violin plots were performed as Dunn test with Bonferroni correction. **d**. Heatmap comparing the CUT&Tag signals for H4K16ac and H3K9me3 for WT and MSL1 KO samples across L1 subfamilies, LTRs and ERV subfamilies, Alu subfamilies and NCBI refseq genes.

**a**

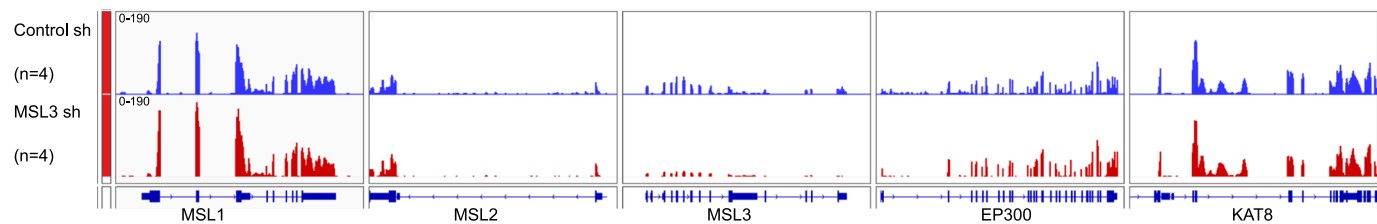

**b**

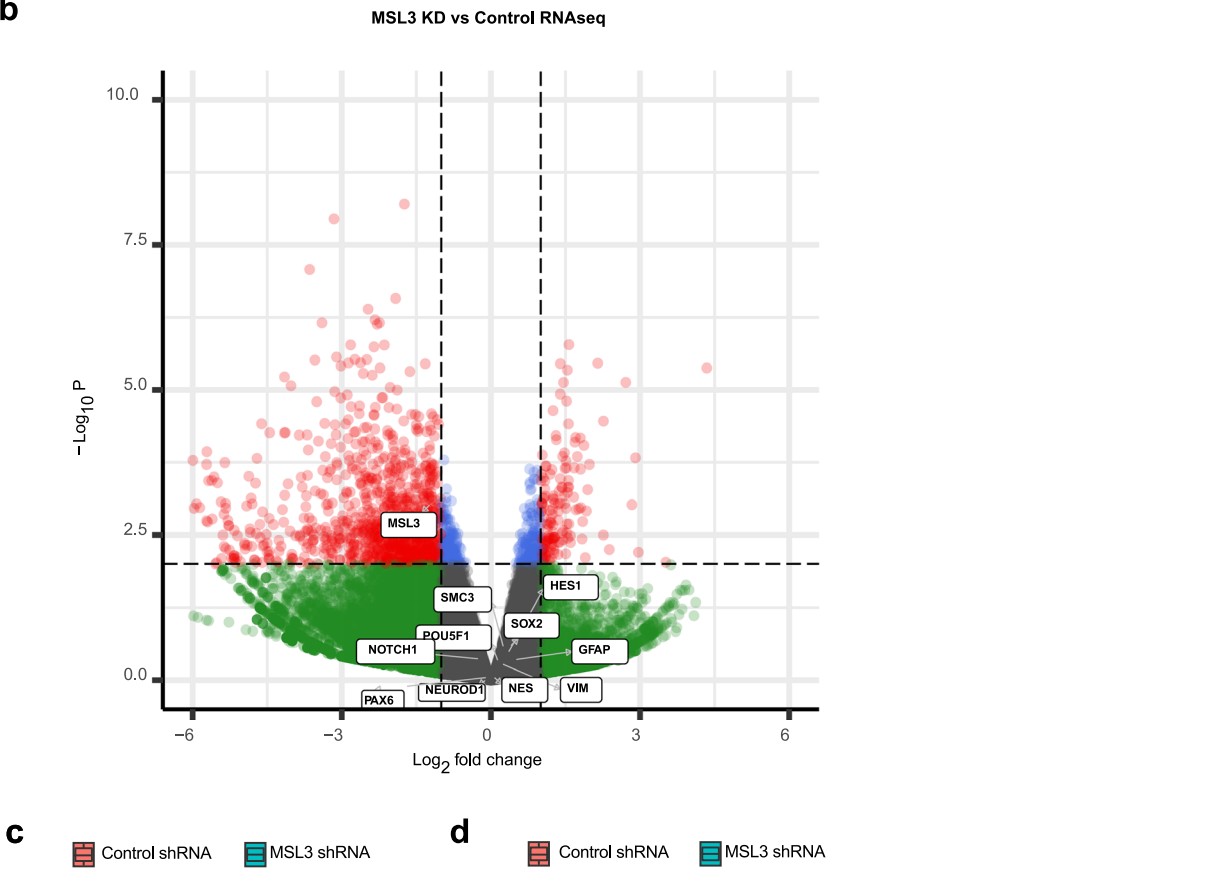

**c**　　　　**d**

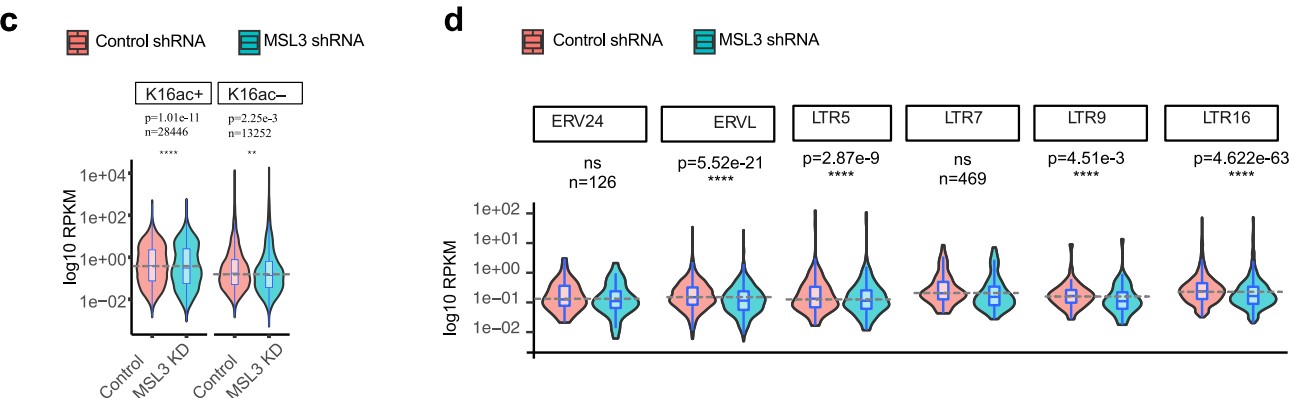

**Extended Data Fig. 8 | Related to Fig. 5, MSL3 depletion data. a.** IGV browser tracks showing RNAseq reads (RPKM) at *MSL1, MSL2, MSL3* and *KAT8* locus in control knockdown (nontargeting shRNA) and MSL3 knockdown (n = 4, biological replicates) H9 hESCs. **b.** Volcano plot showing up-and down-regulated genes upon lentiviral shRNA mediated knockdown of MSL3. Pluripotency-associated genes (for example, POU5F1, NANOG, SOX2) and genes expressed in neuronal differentiation (for example, PAX6, GFAP, NES, NEUROD1) are shown in arrow marks. **c.** Violin plots for the RNAseq signal (log10 RPKM) for the Control-shRNA and MSL3-shRNA knockdown inH9 hESCs for genes that contain H4K16ac peak (H4K16ac+) and genes that lack H4K16ac peaksor (H4K16ac−). **d.** Like C but for LTR subfamilies (ERV24, ERVL, LTR5, LTR7, LTR9 and LTR16; bottom panel).

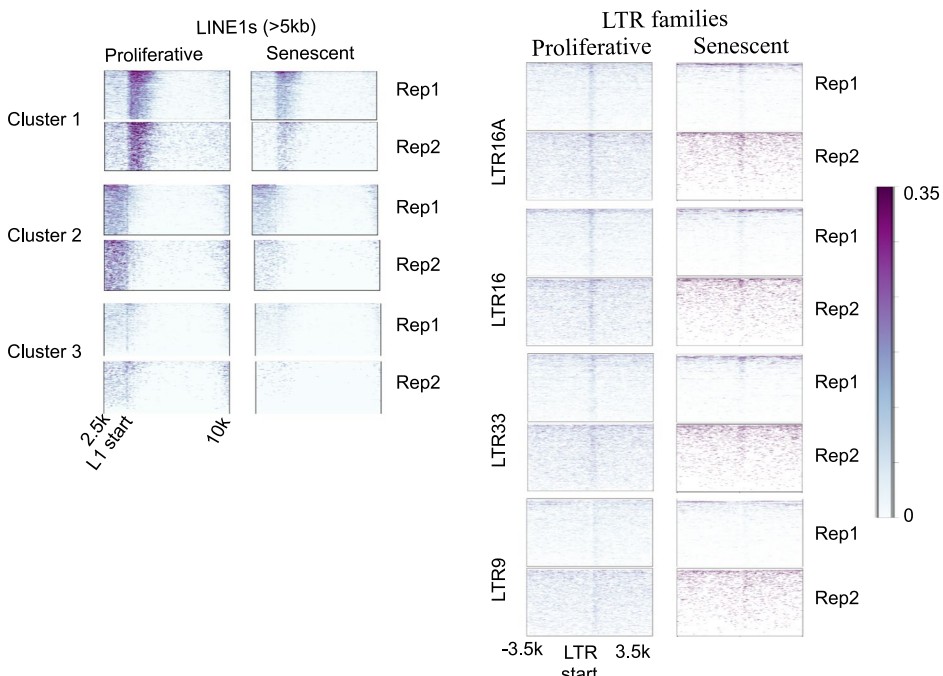

**Extended Data Fig. 9 | H4K16ac is enriched at TEs in proliferative cells compared to senescent cells.** H4K16ac ChIPseq/input signal for L1 (left) and LTR (right) subfamilies in the proliferative and senescent IMR90 cell line.

# Reporting Summary

## Statistics

For all statistical analyses, confirm that the following items are present in the figure legend, table legend, main text, or Methods section.

| n/a | Confirmed | |
|---|---|---|
| ☐ | ☒ | The exact sample size (*n*) for each experimental group/condition, given as a discrete number and unit of measurement |
| ☐ | ☒ | A statement on whether measurements were taken from distinct samples or whether the same sample was measured repeatedly |
| ☐ | ☒ | The statistical test(s) used AND whether they are one- or two-sided<br>*Only common tests should be described solely by name; describe more complex techniques in the Methods section.* |
| ☒ | ☐ | A description of all covariates tested |
| ☐ | ☒ | A description of any assumptions or corrections, such as tests of normality and adjustment for multiple comparisons |
| ☐ | ☒ | A full description of the statistical parameters including central tendency (e.g. means) or other basic estimates (e.g. regression coefficient) AND variation (e.g. standard deviation) or associated estimates of uncertainty (e.g. confidence intervals) |
| ☐ | ☒ | For null hypothesis testing, the test statistic (e.g. *F*, *t*, *r*) with confidence intervals, effect sizes, degrees of freedom and *P* value noted<br>*Give P values as exact values whenever suitable.* |
| ☒ | ☐ | For Bayesian analysis, information on the choice of priors and Markov chain Monte Carlo settings |
| ☒ | ☐ | For hierarchical and complex designs, identification of the appropriate level for tests and full reporting of outcomes |
| ☒ | ☐ | Estimates of effect sizes (e.g. Cohen's *d*, Pearson's *r*), indicating how they were calculated |

*Our web collection on statistics for biologists contains articles on many of the points above.*

## Software and code

Policy information about availability of computer code

| Data collection | No softwares was used for data collection |
|---|---|
| Data analysis | RepeatMasker https://www.repeatmasker.org v4.0.7<br>Bowtie2 http://bowtie-bio.sourceforge.net/bowtie2/ v2.4.5<br>STAR https://github.com/alexdobin/STAR v2.7.0f<br>SAMtools http://www.htslib.org/ v1.10<br>BEDTools https://github.com/arq5x/bedtools2/ v2.28.0<br>deepTools https://github.com/deeptools/deepTools v3.5.1<br>ChromHMM http://compbio.mit.edu/ChromHMM/<br>ComplexHeatmap https://jokergoo.github.io/ComplexHeatmap/ v2.10.0<br>DESeq2 https://github.com/mikelove/DESeq2/ v1.34.0<br>Intervene https://github.com/asntech/intervene v0.6.5<br>SEACR https://github.com/FredHutch/SEACR v1.3<br>Trimmomatic v0.36<br>GraphPad Prsim v9 |

For manuscripts utilizing custom algorithms or software that are central to the research but not yet described in published literature, software must be made available to editors and reviewers. We strongly encourage code deposition in a community repository (e.g. GitHub). See the Nature Portfolio guidelines for submitting code & software for further information.

## Data

Policy information about <u>availability of data</u>

All manuscripts must include a <u>data availability statement</u>. This statement should provide the following information, where applicable:

- Accession codes, unique identifiers, or web links for publicly available datasets
- A description of any restrictions on data availability
- For clinical datasets or third party data, please ensure that the statement adheres to our <u>policy</u>

All the NGS data generated in this study can be accessed at NBCI GEO under the accession ID GSE200770.
CUT&Tag, RNAseq and ATACseq datasets are deposited in the NCBI GEO datasets with following accession IDs.
CUT&Tag GSE200768
RNAseq GSE200769
ATACseq GSE200767
Details of the datasets used from other studies are mentioned in the Supplimentary Table S1.
Genome assemblies used to map the data were hg38 for human and mm10 for mouse.

## Human research participants

Policy information about <u>studies involving human research participants and Sex and Gender in Research.</u>

| | |
|---|---|
| Reporting on sex and gender | N/A |
| Population characteristics | N/A |
| Recruitment | N/A |
| Ethics oversight | N/A |

Note that full information on the approval of the study protocol must also be provided in the manuscript.

# Field-specific reporting

Please select the one below that is the best fit for your research. If you are not sure, read the appropriate sections before making your selection.

☒ Life sciences          ☐ Behavioural & social sciences          ☐ Ecological, evolutionary & environmental sciences

For a reference copy of the document with all sections, see <u>nature.com/documents/nr-reporting-summary-flat.pdf</u>

# Life sciences study design

All studies must disclose on these points even when the disclosure is negative.

| | |
|---|---|
| Sample size | Standard number of biological replicates and sequencing depth. For CUT&Tag, two or three biological replicates were used. For ATACseq three biological replicates were used and for RNAseq two biological replicates were used following the similar studies. For all RTqPCR experiments, 3 independent experiments were performed. For western blotting and microscopy, representative data from 2 independent experiments have been shown. |
| Data exclusions | No data is excluded from analyses |
| Replication | CUT&Tag, ATACseq and RNAseq experiments are replicated 2-3 times. For all RTqPCR experiments, 3 independent experiments were performed. For western blotting and microscopy, representative data from 2 independent experiments have been shown. |
| Randomization | Randomization for the samples was not required as a part of this study, as the experimentation was mainly on cell lines . |
| Blinding | Not applicable for this study |

# Reporting for specific materials, systems and methods

We require information from authors about some types of materials, experimental systems and methods used in many studies. Here, indicate whether each material, system or method listed is relevant to your study. If you are not sure if a list item applies to your research, read the appropriate section before selecting a response.

## Materials & experimental systems

| n/a | Involved in the study |
|---|---|
| ☐ | ☒ Antibodies |
| ☐ | ☒ Eukaryotic cell lines |
| ☒ | ☐ Palaeontology and archaeology |
| ☒ | ☐ Animals and other organisms |
| ☒ | ☐ Clinical data |
| ☒ | ☐ Dual use research of concern |

## Methods

| n/a | Involved in the study |
|---|---|
| ☐ | ☒ ChIP-seq |
| ☒ | ☐ Flow cytometry |
| ☒ | ☐ MRI-based neuroimaging |

# Antibodies

| | |
|---|---|
| Antibodies used | Primary antibodies for CUT&Tag:<br>Normal rabbit IgG (Santa Cruz, sc-2027)<br>H3K27ac (Abcam, ab4729)<br>H4K16ac (Abcam, ab109463, Clone number EPR1004)<br>H3K122ac (Abcam, ab33309)<br>H3K4me1(Abcam, ab8895)<br>H3K36me3 (Abcam, ab9050)<br>H3K4me3 (Millipore, 07-473)<br>H3K27me3 (Abcam, ab192985, Clone number EPR18607)<br>H3K9me3 (Abcam, ab176916, Clone number EPR16601)<br>1 µg of primary antibodies was added to samples for CUT&Tag experiments and incubated at 4ºC overnight in a nutator.<br>Secondary antibody for CUT&Tag experiments:<br>Guinea pig α-rabbit antibody (Antibodies online, ABIN101961, 1:100 dilution)<br><br>Primary antibodies for Western blotting:<br>MSL3 (Merck Millipore, ABE467, 1:1000 dilution)<br>L1 ORF1 (Merck Millipore, MABC1152, Clone 4H1, 1:1000 dilution)<br>H4K16ac (Abcam, ab109463, Clone number EPR1004, 1:5000 dilution)<br>H3K27ac (Abcam, ab4729, 1:5000 dilution)<br>α-tubulin (Sigma, T9026, Clone DM1A, 1:5000 dilution)<br>HERV (Novus Biologicals, NB100-93579, 1:500 dilution)<br>Secondary antibody for Western blotting:<br>Goat anti-rabbit IgG H&L HRP (Abcam, ab6721, 1:3000-1:10,000 dilution)<br>Goat anti-mouse H&L HRP (ThermoFisher Scientific, 31430, 1:3000-1:10,000 dilution) |
| Validation | 3 different H4K16ac antibodies from commercial vendors (Abcam, Millipore and Cell signalling technologeis) were tested for CUT&Tag. All three antibodies showed similar profile. Furthermore, western blotting, immunofluorescence (Fig 2) and CUT&Tag (Extended data Fig 4) upon MSL knockout cell lines showed significant reduction in H4K16ac CUT&Tag signal confirming the specificity of the antibodies.<br><br>Commercial antibodies, L1 ORF1, H3K27ac, H4K12ac, H3K9me3, H3K4me1, H3K4me3 antibodies are validated for western blot and ChIP. The references are available in the manufacturer's website. H3K122ac antibodies are validated in Tropberger et al Cell 2013 and Pradeepa et al 2016 Nature Genetics.<br>α-tubulin and HERV antibodies validated for western blot and ChIP. The references are available in the manufacturer's website. |

# Eukaryotic cell lines

Policy information about cell lines and Sex and Gender in Research

| | |
|---|---|
| Cell line source(s) | H9 female hESC is procured from WiCell.<br>TDF iCAS9 lines are from Dr. Paola Scaffidi's lab (Crick Institute).<br>H1 male hESC iCAS9 lines are from Dr. Silvia Santos's lab (Crick Institute)<br>Prostate cancer lines (LNCaP, PC3, RWPE) are from ATCC.<br>HEK293T line (ATCC) is from Prof. Inderjeet Dokal's lab (Blizard Institute).<br>K562 (ATCC) is from Dr. Miguel Branco's lab (Blizard Institute).<br>SH-SY5Y (ATCC) cell line is from Dr. Elena Bochukova's lab (Blizard Institute). |
| Authentication | Cell lines are not authenticated |
| Mycoplasma contamination | Cell lines are routinely tested for mycoplasma contamination, cell lines were negative for mycoplama before performing the experiments |

| Commonly misidentified lines (See ICLAC register) | No commonly misidentified cell lines used in the study |
|---|---|

# ChIP-seq

## Data deposition

☒ Confirm that both raw and final processed data have been deposited in a public database such as GEO.

☒ Confirm that you have deposited or provided access to graph files (e.g. BED files) for the called peaks.

| Data access links *May remain private before publication.* | The data discussed in this publication have been deposited in NCBI's Gene Expression Omnibus (Pal et al., 2023) and are accessible through GEO Series accession number GSE200770 (https://www.ncbi.nlm.nih.gov/geo/query/acc.cgi?acc=GSE200770). All the datasets generated and used in this study are detailed in Supplementary Table 1. |
|---|---|
| Files in database submission | Raw Fastq files, peak files and bigwig files |
| Genome browser session (e.g. UCSC) | HG38 genome browser session links<br><br>https://data.cyverse.org/dav-anon/iplant/home/pradeepam/H9_CutandTag_Merged_UniqueMapped/H3K27ac_merged_unq_coverage.bw<br>https://data.cyverse.org/dav-anon/iplant/home/pradeepam/H9_CutandTag_Merged_UniqueMapped/H3K27me3_merged_unq_coverage.bw<br>https://data.cyverse.org/dav-anon/iplant/home/pradeepam/H9_CutandTag_Merged_UniqueMapped/H3K36me3_unq_coverage.bw<br>https://data.cyverse.org/dav-anon/iplant/home/pradeepam/H9_CutandTag_Merged_UniqueMapped/H3K4me1_merged_unq_coverage.bw<br>https://data.cyverse.org/dav-anon/iplant/home/pradeepam/H9_CutandTag_Merged_UniqueMapped/H3K4me3_merged_unq_coverage.bw<br>https://data.cyverse.org/dav-anon/iplant/home/pradeepam/H9_CutandTag_Merged_UniqueMapped/H4K12ac_merged_unq_coverage.bw<br>https://data.cyverse.org/dav-anon/iplant/home/pradeepam/H9_CutandTag_Merged_UniqueMapped/H4K16ac_merged_unq_coverage.bw<br>https://data.cyverse.org/dav-anon/iplant/home/pradeepam/H9_CutandTag_Merged_UniqueMapped/H9WT_K122ac_merged_unq_srt_adj.bw<br>https://data.cyverse.org/dav-anon/iplant/home/pradeepam/H9_CutandTag_Merged_UniqueMapped/H9WT_K9me3_merged_unq_srt_adj.bw<br>https://data.cyverse.org/dav-anon/iplant/home/pradeepam/H9_CutandTag_Merged_UniqueMapped/Rb_IgG_merged_unq_coverage.bw |

## Methodology

| Replicates | CUT&Tag, ATACseq and RNAseq experiments are replicated 2-3 times. Replicates agree with each other. |
|---|---|
| Sequencing depth | Sample Total read pairs (post trimming)<br>H9, H3K27ac Rep1 5267795<br>H9, H3K27ac Rep2 24283828<br>H9, H3K27ac Rep3 21041964<br>H9, H3K27me3 Rep1 1059518<br>H9, H3K27me3 Rep2 2330065<br>H9, H3K4me1 Rep1 8726469<br>H9, H3K4me1 Rep2 16756188<br>H9, H3K4me1 Rep3 35390485<br>H9, H3K4me3 Rep1 6822949<br>H9, H3K4me3 Rep2 40759484<br>H9, H3K4me3 Rep3 1452386<br>H9, H4K12ac Rep1 7805697<br>H9, H4K12ac Rep2 23829711<br>H9, H4K12ac Rep3 27318999<br>H9, H4K16ac Rep1 8831804<br>H9, H4K16ac Rep2 18212861<br>H9, H4K16ac Rep3 21797848<br>H9, H3K122ac Rep1 1434855<br>H9, H3K122ac Rep2 2802275<br>H9, H3K9me3 Rep1 16787273<br>H9, H3K9me3 Rep2 1776891<br>H9, IgG Rep1 1455306<br>H9, IgG Rep2 1000089<br>H9, IgG Rep3 10973836 |
| Antibodies | Primary antibodies for CUT&Tag:<br>Normal rabbit IgG (Santa Cruz, sc-2027) |

H3K27ac (Abcam, ab4729)
H4K16ac (Abcam, ab109463,  Clone number EPR1004)
H3K122ac (Abcam, ab33309)
H3K4me1(Abcam, ab8895)
H3K36me3 (Abcam, ab9050)
H3K4me3 (Millipore, 07-473)
H3K27me3 (Abcam, ab192985,  Clone number EPR18607)
H3K9me3 (Abcam, ab176916,  Clone number EPR16601)
1  µg of primary antibodies was added to samples for CUT&Tag experiments and incubated at 4ºC overnight in a nutator.
Secondary antibody for CUT&Tag experiments:
Guinea pig α-rabbit antibody (Antibodies online, ABIN101961, 1:100 dilution)

**Peak calling parameters**

CUT&Tag peaks were called using SEACR v1.3 with relaxed (threshold 1e-6) as well as stringent (thareshold 0.01) parameters without IgG noramlisation.

**Data quality**

Peak detection was performed using stringent as well as relaxed mode with stringent cut-off (threshold <=1e-6).

**Software**

Trimmomatic v0.36 to trim the adapters
Bowtie2 http://bowtie-bio.sourceforge.net/bowtie2/ v2.4.5 for mapping to hg38 genome
SAMtools http://www.htslib.org/ v1.10 sorting and indexing bam files
SEACR v1.3 for peak calling, Repeatasker RepeatMasker v4.0.7 for repeat content analyis,
BEDTools v2.28.0 fro peak intersections,
deepTools v3.5.1 fror plotting bigwig signals across genomic landmarks
ChromHMM for genome annotation of peaks and genomic coordinates
Intervene v0.6.5 for peaks overlap across multiple datasets
RepeatMasker https://www.repeatmasker.org  v4.0.7 to obtain repeat content across peaks

