## [Peer Review File · Nature Structural & Molecular Biology]

Peer Review Information

Manuscript Title: H4K16ac activates the transcription of transposable elements and contributes to their cis regulatory function

Corresponding author name(s): Madapura Pradeepa

Reviewer Comments & Decisions:

Decision Letter, initial version:
--

Message: Dear Dr. Pradeepa,

Thank you for submitting your manuscript "H4K16ac activates the transcription of transposable elements and contributes to their cis regulatory function". I apologize for the delay in processing your manuscript, which resulted from difficulties in obtaining referees' reports. Nevertheless, the comments from the 4 reviewers who have evaluated your manuscript are below. Unfortunately, after carefully considering their comments, we cannot offer to publish your manuscript in Nature Structural & Molecular Biology.

You will see that while the referees find the work of potentially interesting, they raise concerns about the strength of the novel conclusions that can be drawn at this stage, as well as concerns about the robustness of the data and the analysis performed.

However, if further experimentation, analysis, and revisions allow you to address the referees concerns in full, we would be prepared to consider an appeal of our decision, on the condition that no related work is published in the interim or has been accepted in our journal. Please contact me to discuss an appeal and potential revision. Please note that, until we have the opportunity to read the revised manuscript in its entirety, we cannot promise that it will be sent back for peer review.

I am sorry we could not be more positive on this occasion. I hope that you find the referees' comments useful in deciding how best to proceed.

Sincerely,

Carolina

Carolina Perdigoto, PhD
Chief Editor
Nature Structural & Molecular Biology
orcid.org/0000-0002-5783-7106

Reviewers' Comments:

Reviewer #1:

Remarks to the Author:

Pal and Patel et al have carried out global characterisation of histone marks and other chromatin features known to be associated with active enhancers in human embryonic stem cells, cancer cell lines and tissues using publically available datasets and generating original data using CUT&TAG and ATACseq. They specifically observed enrichment of H4K16ac and H4K122ac on transposable elements, mediated by the MSL complex. Furthermore, they have shown that H4K16ac marked TEs act as cis enhancers and regulate neighbouring gene expression using forefront techniques like CRISPRi and that these lie on TAD borders influencing genome architecture using HiC. Overall, data is well presented and provides novel insight into the role of TEs in shaping our genome but the manuscript will benefit from addressing the following issues:

1. Introduction should be detailed further in order to include literature related to known chromatin features associated with TEs such as DNA methylation and H3K9me3 mark, known transcription factors and other regulatory proteins.
2. Results section will become much easier to follow by the readers if subheadings are introduced.
3. Since H3K9me3 is the most enriched mark on L1s, it is worth exploring and commenting on its relationship with H4K16ac and H4K122ac. Looking at Fig 1F the marks are exclusive to each other, which is expected as these are associated with opposite chromatin states. Still including the heatmap of H3K9me3 in Fig 1E and then discussing the result will be useful. Basically, out of total full length L1s how many belong to the potential enhancer category?
4. In the heatmap of Fig 4A, p values should be included.
5. To validate specificity and enhancer function of L1s or LTRs marked with H4K16ac and H4K122ac, CRISPRi on unmarked TEs should be checked. A previous publication from one of the authors: Functional evaluation of transposable elements as enhancers in mouse embryonic and trophoblast stem cells, eLife 2019 should be cited and the chromatin marks on previously identified enhancer and nonenhancer TEs should be explored and discussed.
6. Since authors are exploring data of different cell types, it would be interesting to comment and discuss tissue specificity observed amongst the L1s or LTRs marked with H4K16ac and H4K122ac.
7. Finally, correlation between DNA methylation and the histone marks (H4K16ac and H4K122ac) should be explored. For example, upon MSL knockdown does TEs gain methylation or only loss of the active histone marks sufficient to downregulate expression?

Thanks and best wishes

Reviewer #2:

Remarks to the Author:

In this work, Pal and colleagues investigated the histone modification in human embryonic stem cells and several somatic cells, and found the H4K16ac enriched in the 5'UTR of L1 retrotransposons. They knockdown MSL3 with shRNA to show the H4K16ac activates the transcription of those TEs. They performed CRISPRi to show the TEs function as enhancers to regulate genes in cis.

In general, I am quite concerned about the novelty of this manuscript, many of their findings are known but lack significant advances in the area. It is known that L1 5'UTR enriches many histone modifications including both acetylation and methylation and acts as enhancers. Besides, the lack of necessary quality control leads that the data do not fully support the conclusions drawn at this stage.

Major points

1. In Fig.1F, only H4K16ac and H3K122ac are enriched while depletion of other modifications. However, this is most likely wrong, it is known that L1 5'UTR enriched many histone modifications, especially for H3K9me3 (just as shown in Fig.1B and Fig3A). Moreover, the INPUT/IgG control needs to show for Fig1E, F.
2. For Fig2D, as shRNA is often toxic and has many side effects, rescue experiments are required. Furthermore, are the putative target genes downregulated after loss of MSL3? Statistic results from the RNA-seq data will be helpful to address whether those TEs can regulate gene expression.
3. Fig4A-C, the HiC data are too sparse to support the authors' claim. For Fig4D, a schematic plot like Fig.4C to show the TE loci are targeted by dCas9 will be helpful for readers to understand how they choose those sites. Meanwhile, the cross negative control is required to show the gene expression is exclusively regulated by its own TE-derived cis-elements but not others to exclude any side effects.
4. In the method part, the authors show that multi-mapped reads were filtered out, and only uniquely mapped reads were retained for all the analysis. However, this approach should be avoided as it tends to greatly underestimate or even eliminate the signal associated with young TE families. This issue has been extensively discussed by many previous studies (Elsasser et al., 2015, Nature; Liu et al., 2021, Nature; Lanciano et al., 2020, Nature Reviews). This is especially important for the young primate-specific L1PAs and human-specific L1Hs which are focused on in this manuscript.

Reviewer #3:

Remarks to the Author:

In this paper, the authors set out to demonstrate that MSL/H4K16ac activate transcription of L1 and ERV/LTR families, and H4K16ac and H3K122ac contribute to the cis-regulatory activity of human and primate-specific L1 and HERVs. Using hESCs, the authors reported H4K16ac and H3K122ac occur at L1 and LTR families and that these marks are associated with 3D genome marks, such as CTCF and TAD borders. Following, they disturbed MSL protein to reduce H4K16ac, where they found downregulation of TEs and associated genes. They also did CRISPRi to demonstrate reduction in gene expression with repression of H4K16ac and H3K122ac L1 and HERVs.

This manuscript contains some very interesting data. However I have some concerns with the quality of the data, as well as with data interpretation.

1. It is not clear how the authors decided to focus on L1 and ERV. Fig 1C suggests higher fraction overlapping Alus. Figure 3 reports a lot of histone modifications co-localized with H4K16ac. By knocking down MSL3, there's a global down-regulation of TE expression including genes from figure 2B and 2E, and a global depletion of H4K16ac across the genome. The co-localization of H4k16ac and H3k9me3 is interesting; there seems to be an intriguing anti-correlation between H4k16ac and BRD4. However, none of the data establishes a causal link between H4K16ac and L1 expression/function regulation. The authors did not fully address the meaning of these co-localization patterns.
2. The study is mostly based on analyzing cut&tag data of various histone marks. Thus, quality of such data is critical. Given guidelines provided by projects such as ENCODE, the authors should provide a complete set of quality assurance, for example, how many peaks were called from each replicate? What is the fraction that is reproducible between replicates? Many of the browser views provided throughout the paper and supplements reveal poor quality of data. Replicates should always be included. Peaks called in each dataset should be clearly labeled. Numbers of peaks from each replicate should be clearly described.
3. Key statistics that describe the relationship between H4k16ac and TEs are not clearly described. For example, what is the fraction of reproducible H4K16ac peaks overlapping TEs? This number is not evident from Figure 1A-C. What is the fraction of each TE families (L1s, ERVs, ALUs, etc) overlapping H4K16ac peaks?
4. Figure 1E suggests that the vast majority of full length L1s seem to have 5' H4K16ac peaks. But the transcription status of these L1s is not shown. How many of these TEs are actually being expressed? Similarly, the transcription status of TEs in Fig 2D/E should be analyzed.
5. The authors claimed: "Notably, L1 and LTR regions that overlap with H4K16ac are particularly downregulated in MSL1 KO cells compared to L1 and LTRs that lack H4K16ac (Fig. 2H)". However, this seems to be true only for LTRs, not L1s, according to Fig. 2H. The expression of K16ac- vs K16ac+ genes should also be included into comparison. Also, the number of elements in each category should be labeled: How many K16ac- L1s are there? Since the vast majority of L1s seem to have H4K16ac (Figure 1E).
6. Figure 2 has many additional issues. 2E description says genes "minimal" but has indication of being statistically significant; 2D seems to have different color scales but there is no explanation; 2F does not have scale bar.
7. Fig4a - "architectural proteins CTCF, RAD21, and their cofactor ZNF143 were enriched at H4K16ac- rather than H3K27ac- marked TEs"- is not supported by data (CTCF-enrich in L1, LTR, RAD21- enriched in Alu, ZNF143- enriched in Alu, LTR).
8. L1 related enhancer analysis needs to include both positive and negative controls.

Reviewer #4:

Remarks to the Author:

Summary: H3K122ac/H4K16ac has recently been shown to mark enhancers, often in the absence of H3K27ac which has historically been the marquee histone PTM defining active enhancers. In this paper, the authors show that the H3K122ac/H4K16ac marks can be found at L1 and ERV sequences and that these marks are correlated with their expression, when the mark is depleted it results in downregulation of transcription, and these TEs can putatively act as regulatory elements for neighboring genes.

Originality: the result of H3K122ac/H4K16ac marking enhancers was shown in Pradeepa, et al. Nature Genetics 2016 and Taylor, et al. Genome Research 2013, both of which are

excellent papers that the senior author of this manuscript was an author on. Many reports have already been published indicating that TE can act as regulatory elements (in human and mouse pluripotent cells, see Fuentes, et al. ELife 2018, and MacFarlan, et al. Nature 2012). Thus both of the key results of this manuscript are not novel.

Data and methodologies: mostly appropriate, but would urge the authors to improve on the quantitative depiction of their genomic/epigenomic data (missing scale bars for heatmaps, y-axis scales, etc pervasively throughout the manuscript).

Conclusions: the most important test of their observations is Fig 4D, in which they target TE and test if chromatin repression using the Crispr-I strategy results in decreased putative target gene expression. This experiment shows that 6/7 of the genes have lower expression levels after dCas9-KRAB targeting of the TEs—and the effect size is comparable to what we normally expect with this technologies' dynamic range. However, since the gRNA likely target hundreds or thousands of sites across the genome since L1 and ERV are typically high copy # and similar in DNA sequence, it would preferable to genetically delete several of these putative enhancers using a genetically clean KO strategy.

References in general are appropriately cited.

Clarity of text: in general I was able to follow the authors reasoning of experimental logic and interpretation.

Specific comments to the authors

Fig 1 B: ambiguous what the y-axis means for this stacked barplot. Is the height of the bar indicative of the enrichment over background? Or is the top of the colored segment the enrichment level?

Fig 1E: no scale bar on the heatmap. Is this log₂ FC or FPKM?

Fig 1F: No y-axis scale on the genome browser snapshot. Is this FPKM or log₂ FC? Are all of the browser tracks all on the same scale?

Fig 2C: Western blot should indicate what HERV protein is being detected, ie HERVK, HERVH, HERV-W etc and gag, env, capsid etc. Also would be more appropriate if the western blot showed a histone loading control, preferably pan-H3—since it looks like knock-down of MSL3 leads to an increase in H3K27ac.

The authors note in Fig 2E that the effect of MSL3 on gene expression was minimal, but the figure panel is labeled with “****” indicating that this effect is highly significant statistically. To clarify this statement, maybe the authors would like to amend that statement to indicate that “the fold/change or effect size on gene expression was minimal”.

Fig 2D: the effect of MSL3 depletion on ATAC-seq levels should be quantified with a statistical test, not just depicted as heatmap. Typically, we would depict a comparison such as control vs knockdown as a log₂Fold-change of the experimental/control—it is unclear based on the labeling of the figure legend and the panel if this is a subtraction (control minus experimental) operation and why the authors would chose to display these data in such a non-standard way. There is also no heatmap scale bar on this figure panel—so it is difficult to interpret the absolute vs relative levels of these data. The authors might want to plot this panel using substantively different colors from dark purple

and dark blue to help guide the reader that they are viewing Cut-N-tag data vs ATAC-seq respectively.

Fig 2 in general: typical to indicate the n=sample size for each genome feature or category, this can be listed in the figure legend or directly on the figure panel itself, but is not NSMB formatting practice to make the reader refer to a supplemental statistics table to obtain this information.

Fig 3: no scale bar on the heatmaps.

Fig 3E: no y-axis scale for these data

Fig 3E: LTR subtypes should be indicated. For instance in 3E, is this an LTR5, LTR5A, LTR5B, LTR5HS?

Fig 4D: The Crispr-I technique (using a dCas9-KRAB) fusion is an alternative to genetically deleting the putative enhancer (by using unique sequence around the repeat element to specifically delete the L1 or ERV)—and certainly worth trying as a first pass to screen for putative TE enhancer sequences that indeed regulate the cognate gene. However, the effect size is rather small for several of the analyzed genes. How many total putative TE-enhancer: gene pairs were analyzed? Does Fig 4D depict all of the candidates tested in this assay or is this a subset? I would urge the authors to pick several of these candidate regions and validate them using crispr/cas9 KO to delete the TE genetically. I understand that with primed hESC, making clonal cell lines can be time consuming and technically difficult, but since the authors have chosen to use a Crispr-I strategy, they are globally targeting many individual instances of L1 or ERVs and it is possible that the effect they see on gene expression of putative cognate genes is either indirect or is blunted by having to recruit the dCas9-KRAB fusion to thousands of sites across the genome, thus giving a rather weak effect size.

The evidence that ERVs or L1 can act as enhancers based on the data they present here is rather weak. For instance, chromatin-looping or chromatin-conformation data (such as MicroC) could help boost evidence to support the claim that L1 can act as enhancers of the Moxd1 gene. In the current form, Fig 4C is essentially un-interpretable as there is no quantification of the interaction strength between the L1 and the Moxd1 gene. Is there a genome-wide analysis of these data that can globally support their claims re: TE-enhancers and genes?

Since intronic L1 in the host gene can have a strong effect on the expression of the CYP3a5 gene (see Liu, et al. Nature 2018 which the authors cite) by recruiting the HUSH repressive complex that also decreased the host gene expression or conversely is the case of this manuscript allowing for MSL recruitment boost expression of genes near H3K122ac/H4K16ac marked L1/ERVs, the effect of TE on neighboring genes can be difficult to dissect—and are not necessarily a result of TE-comprised enhancers.

** For Springer Nature Limited general information and news for authors, see <http://npg.nature.com/authors>.

Author Rebuttal to Initial comments

Reviewers' Comments:

Reviewer #1:

Remarks to the Author:

Pal and Patel et al have carried out global characterisation of histone marks and other chromatin features known to be associated with active enhancers in human embryonic stem cells, cancer cell lines and tissues using publically available datasets and generating original data using CUT&TAG and ATACseq. They specifically observed enrichment of H4K16ac and H4K122ac on transposable elements, mediated by the MSL complex. Furthermore, they have shown that H4K16ac marked TEs act as cis enhancers and regulate neighbouring gene expression using forefront techniques like CRISPRi and that these lie on TAD borders influencing genome architecture using HiC. Overall, data is well presented and provides novel insight into the role of TEs in shaping our genome but the manuscript will benefit from addressing the following issues:

1. Introduction should be detailed further in order to include literature related to known chromatin features associated with TEs such as DNA methylation and H3K9me3 mark, known transcription factors and other regulatory proteins.

We have now added more details to the introduction, such as TFs, and other regulatory mechanisms for repressing TEs (first paragraph).

2. Results section will become much easier to follow by the readers if subheadings are introduced.

Thanks for this suggestion, subheadings are added in this version.

3. Since H3K9me3 is the most enriched mark on L1s, it is worth exploring and commenting on its relationship with H4K16ac and H4K122ac. Looking at Fig 1F the marks are exclusive to each other, which is expected as these are associated with opposite chromatin states. Still including the heatmap of H3K9me3 in Fig 1E and then discussing the result will be useful. Basically, out of total full length L1s how many belong to the potential enhancer category?

The reviewer is absolutely correct that H3K9me3 is enriched at many L1 5' UTRs in hESCs. Since H4K16ac and H4K122ac are also enriched at many L1 5' UTRs, some of the L1s have both these marks. However, new analysis of H3K9me3 and H4K16ac peaks at L1s shows anticorrelation between these acetylations and H3K9me3 (new fig. S1C).

4. In the heatmap of Fig 4A, p values should be included.

Since there are too many TFs in the figure, instead we have now plotted the enrichment profile for key TFs at TEs (Fig 3B).

5. To validate specificity and enhancer function of L1s or LTRs marked with H4K16ac and H4K122ac, CRISPRi on unmarked TEs should be checked.

We have now included CRISPRi data for two unmarked L1s in Fig. 4H and J.

A previous publication from one of the authors: Functional evaluation of transposable elements as enhancers in mouse embryonic and trophoblast stem cells, eLife 2019 should be cited and the chromatin marks on previously identified enhancer and nonenhancer TEs should be explored and discussed.

The eLife 2019 paper is now discussed, and chromatin marks at TE enhancers and are mentioned in the introduction (Page 16).

6. Since authors are exploring data of different cell types, it would be interesting to comment and discuss tissue specificity observed amongst the L1s or LTRs marked with H4K16ac and H4K122ac.

We agree and have now included a detailed discussion about tissue specific regulation of TEs by H4K16ac in this version (Page 16)

7. Finally, correlation between DNA methylation and the histone marks (H4K16ac and H4K122ac) should be explored. For example, upon MSL knockdown does TEs gain methylation or only loss of the active histone marks sufficient to downregulate expression?

Even if there is any change in DNA methylation levels in MSL knockdowns, it will be a consequence of transcription repression. It may not be due to direct effect of MSL/H4K16ac depletion, so we believe it is beyond the scope of this work.

Reviewer #2:

Remarks to the Author:

In this work, Pal and colleagues investigated the histone modification in human embryonic stem cells and several somatic cells, and found the H4K16ac enriched in the 5'UTR of L1 retrotransposons. They knockdown MSL3 with shRNA to show the H4K16ac activates the transcription of those TEs. They performed CRISPRi to show the TEs function as enhancers to regulate genes in cis.

In general, I am quite concerned about the novelty of this manuscript, many of their findings are known but lack significant advances in the area. It is known that L1 5'UTR enriches many histone modifications including both acetylation and methylation and acts as enhancers.

We disagree with this comment about the novelty. The reviewer is correct in that LTRs have previously been shown to function as enhancers, importantly this has never been demonstrated for L1s. More specifically, we believe the following findings are novel in this study:

- There are many epigenetic repressive mechanisms known for TEs. Here for the first time, we discovered an epigenetic activation machinery that activates transcription of L1 and LTRs.
- This is the first report to demonstrate L1 5'UTRs function as enhancers by multiple complementary approaches, including CUT&Tag for chromatin marks, ATACseq, and analysis of genome-wide enhancer assays (STARRseq).
- This is the first study to show L1s and LTRs with H4K16ac are enriched at TAD borders. In addition, it is also the first study to show that L1 and LTRs with acetylations engage in long-range chromatin loops with genes and regulate their expression.
- Furthermore, we have demonstrated the role of H4K16ac marked L1 and HERV/LTRs in the regulation of genes in cis by a CRISPR dCAS9-based approach and genetic deletion of L1s in highly relevant but difficult to work with human embryonic stem cells.
- We have also included new Hi-C/microC, HiChIP analysis, new CRISPRi controls and importantly CRISPR mediated deletion of LINE1. Importantly, all these approaches support our conclusions and clearly demonstrate a role of H4K16ac+ L1 to function as enhancers to regulate genes in cis.

The lack of necessary quality control leads that the data do not fully support the conclusions drawn at this stage.

In this version, we have reanalysed all our data and presented the data as suggested by the reviewers, and in addition we have also added several exciting new findings. For example, we have included new analyses in Fig 3 and 5, and we have generated new CRISPR deletion data, and more CRISPRi controls, all of which support our conclusions (new Fig. 4, new Fig 5).

Major points

1. In Fig.1F, only H4K16ac and H3K122ac are enriched while depletion of other modifications. However, this is most likely wrong, it is known that L1 5'UTR enriched many histone modifications, especially for H3K9me3 (just as shown in Fig.1B and Fig3A). Moreover, the INPUT/IgG control needs to show for Fig1E, F.

We agree H3K9me3 is specifically enriched at L1 5' UTRs. Our reanalysis and new figures with all the histone modifications and IgG control clearly show enrichment of H4K16ac, H3K122ac and H3K9me3 at L1 5' UTRs. In marked contrast, at the gene promoters, H3K27ac, H4K12ac, and H3K4me3 are enriched (new Fig 1).

2. For Fig2D, as shRNA is often toxic and has many side effects, rescue experiments are required.

In order to rule out downregulation of TEs is due to shRNA side effects, we now show similar findings from inducible CAS9-mediated knockout of MSL1 and MSL3 in independent hESC line (H1) and in transformed fibroblasts. We have also generated two new replicates RNAseq data with new biological replicates of MSL3 knockdown H9 hESCs, and new high-content imaging data showing reduced L1ORF1p upon for inducible genetic knockout of MSL1 in single cell level. This new data clearly demonstrate the downregulation of L1/LTRs upon knockout of MSL1 and MSL3 (New Fig 5 and fig S7).

Furthermore, are the putative target genes downregulated after the loss of MSL3? Statistic results from the RNA-seq data will be helpful to address whether those TEs can regulate gene expression.

Thanks for this suggestion, our new analysis (Fig 5G, H and I) show that genes proximal to H4K16ac+ L1 and LTRs are significantly downregulated upon MSL3 knockdown, while genes that are located further away from these TEs are not affected. Another new analysis from fibroblasts has revealed that genes that are closer to H4K16ac+ L1 and LTRs are expressed at significantly higher level than genes that are located further away from these TEs (>10 kb) (fig S9). This trend is not observed or opposite in L1 and LTRs that lack H4K16ac.

3. Fig4A-C, the HiC data are too sparse to support the authors' claim.

We agree that Hi-C lacks the resolution to call statistically significant chromatin loops between gene regulatory elements. Therefore, we have now used Micro-C data to identify looping interactions with target genes for L1 and LTRs that are enriched with histone acetylations (Fig 3F&G). We show that significantly higher number of TEs loop with genes compared to TEs that lack acetylations. Furthermore, we have used RAD21 HiChIP data to predict putative target genes in our CRISPRi and new CRISPR deletion experiments. While the significantly higher resolution of Micro-C and HiChIP data enables the reliable identification of chromatin loops, HiC data remains ideally suited for analysing TAD border enrichment, as we have performed in Fig 3C.

For Fig4D, a schematic plot like Fig.4C to show the TE loci are targeted by dCas9 will be helpful for readers to understand how they choose those sites.

We have now used schematics along with RAD21 HiChIP data to show the candidate TEs and strategies for choosing their putative target genes (Fig. 4).

Meanwhile, the cross negative control is required to show the gene expression is exclusively regulated by its own TE-derived cis-elements but not others to exclude any side effects.

We have now included more negative controls for CRISPRi, including CRISPRi for L1 that lack detectable acetylations (Fig 4H and J) and new CRISPR deletion data (Fig. 4K).

4. In the method part, the authors show that multi-mapped reads were filtered out, and only uniquely mapped reads were retained for all the analysis. However, this approach should be avoided as it tends to greatly underestimate or even eliminate the signal associated with young TE families. This issue has been extensively discussed by many previous studies (Elsasser et al., 2015, Nature; Liu et al., 2021, Nature; Lanciano et al., 2020, Nature Reviews). This is especially important for the young primate-specific L1PAs and human-specific L1Hs which are focused on in this manuscript.

We agree with the reviewer's point regarding enrichment at young L1s. We have therefore used unique reads when analysing individual TEs or fraction of TE copies. We included multimapping reads for subfamily-level analyses (Fig 5E). All our sequencing reads are 150 bp paired end which is improved the chances of unique mapped reads. However, as expected H3K9me3, H4K16ac, and H3K122ac are enriched at TEs, this has resulted in less uniquely mapped reads retained for these modifications (Mapping statistics are detailed in table S2).

Reviewer #3:

Remarks to the Author:

In this paper, the authors set out to demonstrate that MSL/H4K16ac activate transcription of L1 and ERV/LTR families, and H4K16ac and H3K122ac contribute to the cis-regulatory activity of human and primate-specific L1 and HERVs. Using hESCs, the authors reported H4K16ac and H3K122ac occur at L1 and LTR families and that these marks are associated with 3D genome marks, such as CTCF and TAD borders. Following, they disturbed MSL protein to reduce H4K16ac, where they found downregulation of TEs and associated genes. They also did CRISPRi to demonstrate reduction in gene expression with repression of H4K16ac and H3K122ac L1 and HERVs.

This manuscript contains some very interesting data. However I have some concerns with the

quality of the data, as well as with data interpretation.

We have detailed the quality of CUT&Tag data and improved our data presentation and detailed our interpretation in this version (fig S1 and 2).

1. It is not clear how the authors decided to focus on L1 and ERV. Fig 1C suggests higher fraction overlapping Alus.

We have now discussed the reasoning behind our decision not to focus on functional investigation of Alu elements. As we did not detect an enrichment for enhancer associated chromatin features at Alus (fig. S4), they are not likely to function as enhancers in hESCs (discussed in page 6).

Figure 3 reports a lot of histone modifications co-localized with H4K16ac. By knocking down MSL3, there's a global down-regulation of TE expression including genes from figure 2B and 2E, and a global depletion of H4K16ac across the genome. The co-localization of H4K16ac and H3K9me3 is interesting; there seems to be an intriguing anti-correlation between H4K16ac and BRD4. However, none of the data establishes a causal link between H4K16ac and L1 expression/function regulation. The authors did not fully address the meaning of these co-localization patterns.

In this version, we have discussed the H3K9me3/H4K16ac colocalization at L1s (Page 4, first paragraph), and we have included a new figure (fig S1C). Moreover, we show that depletion of H4K16ac does not alter H3K9me3 levels at TEs in MSL KO fibroblasts (fig S7D).

2. The study is mostly based on analyzing cut&tag data of various histone marks. Thus, quality of such data is critical. Given guidelines provided by projects such as ENCODE, the authors should provide a complete set of quality assurance, for example, how many peaks were called from each replicate? What is the fraction that is reproducible between replicates? Many of the browser views provided throughout the paper and supplements reveal poor quality of data. Replicates should always be included. Peaks called in each dataset should be clearly labeled. Numbers of peaks from each replicate should be clearly described.

In this version, we have detailed the quality of CUT&Tag data, replicates, peaks etc.

We have included individual replicates data in the main and supplementary figures (fig S1 and 2). It is worth noting that H4K16ac ChIP-seq is notoriously difficult to perform in human cell lines. We have had many personal communications from three groups who are experts in chromatin/histone modifications working on NSL, MSL, and H4K16ac, and they have all struggled/failed to perform good quality ChIPseq for this mark in human cell types. Hence, we have spent lot of time optimising and comparing crosslinked ChIP, Native ChIP, CUT&Run and

CUT&Tag for H4K16ac. We found that CUT&Tag works much better compared to other approaches in various human cell lines and tissues. Moreover, when we deplete MSL proteins, we lose H4K16ac at the L1 and LTRs (fig S7D), demonstrating the specificity of our H4K16ac CUT&Tag data.

3. Key statistics that describe the relationship between H4k16ac and TEs are not clearly described. For example, what is the fraction of reproducible H4K16ac peaks overlapping TEs? This number is not evident from Figure 1A-C. What is the fraction of each TE families (L1s, ERVs, ALUs, etc) overlapping H4K16ac peaks?

All the peak statistics are now shown in supplementary tables (table S2 and 3.

4. Figure 1E suggests that the vast majority of full length L1s seem to have 5' H4K16ac peaks. But the transcription status of these L1s is not shown. How many of these TEs are actually being expressed? Similarly, the transcription status of TEs in Fig 2D/E should be analyzed.

Although most L1s have an H4K16ac signal, peak calling detected ~6000 H4K16ac peaks that overlapped with L1 5' UTR out of ~10,000. However, RNAseq data shows that not all H4K16ac marked L1s. More specifically, older L1s do not produce stable and full-length transcripts suggesting that many older H4K16ac marked TEs could function primarily as enhancers or produce short unstable bidirectional transcripts (like enhancer RNAs).

Our new Fig 5 E now includes RNAseq data.

5. The authors claimed: "Notably, L1 and LTR regions that overlap with H4K16ac are particularly downregulated in MSL1 KO cells compared to L1 and LTRs that lack H4K16ac (Fig. 2H)". However, this seems to be true only for LTRs, not L1s, according to Fig. 2H. The expression of K16ac- vs K16ac+ genes should also be included into comparison. Also, the number of elements in each category should be labeled:

We have reanalysed the RNAseq data and presented the data as RPKM from MSL KO fibroblasts (Fig S7B). Our new analysis clearly shows significantly reduced transcription levels of L1s and LTRs that are marked with H4K16ac. In contrast, L1s that lack H4K16ac show no change, while LTRs that do not overlap with H4K16ac also show reduced transcription levels, although less fold change compared to H4K16ac+ LTRs).

We have now compared the H4K16ac+ and H4K16ac- genes in hESCs (figure S8C).

How many K16ac- L1s are there? Since the vast majority of L1s seem to have H4K16ac (Figure 1E).

In hESCs, about 51% of the full-length L1s overlap with reproducible H4K16ac peaks. Now detailed on page 3, last paragraph and (fig. S1C).

6. Figure 2 has many additional issues. 2E description says genes “minimal” but has indication of being statistically significant; 2D seems to have different color scales but there is no explanation; 2F does not have scale bar.

We have replotted these figures with additional two biological replicates of RNAseq. We have reanalysed and presented the detailed gene expression changes (Fig 5 F-H and fig S7 and S8). We have also now added scale bars for all the heatmaps.

7. Fig4a - “architectural proteins CTCF, RAD21, and their cofactor ZNF143 were enriched at H4K16ac- rather than H3K27ac- marked TEs”- is not supported by data (CTCF-enrich in L1,LTR, RAD21- enriched in Alu, ZNF143- enriched in Alu, LTR).

Thanks for spotting this generalised statement, we have now amended this in the text with a more detailed analysis, including a meta-plot showing the enrichment of CTCF, RAD21 and YY1 at L1 and LTRs, and we discuss supporting evidence from the literature (Fig. 3B).

8. L1 related enhancer analysis needs to include both positive and negative controls.

We have now included several additional controls, including L1s with no acetylations and also tested for downregulation of specific target genes that interact in 3D space but not other nearby genes (Fig 4). Furthermore, we have used CRISPR CAS9 to delete 4 H4K16ac+ L1s and assayed their effect on the expression of genes they interact with.

Reviewer #4:

Remarks to the Author:

Summary: H3K122ac/H4K16ac has recently been shown to mark enhancers, often in the absence of H3K27ac which has historically been the marquee histone PTM defining active enhancers. In this paper, the authors show that the H3K122ac/H4K16ac marks can be found at L1 and ERV sequences and that these marks are correlated with their expression, when the mark is depleted it results in downregulation of transcription, and these TEs can putatively act as regulatory elements for neighboring genes.

Originality: the result of H3K122ac/H4K16ac marking enhancers was shown in Pradeepa, et al. Nature Genetics 2016 and Taylor, et al. Genome Research 2013, both of which are excellent papers that the senior author of this manuscript was an author on. Many reports have already been published indicating that TE can act as regulatory elements (in human and mouse

pluripotent cells, see Fuentes, et al. ELife 2018, and MacFarlan, et al. Nature 2012). Thus both of the key results of this manuscript are not novel.

We agree that some of the LTRs subfamilies have previously been demonstrated to function as regulatory elements. However, L1s are not yet demonstrated to function as enhancer elements. However, in this report, we are the first to demonstrate that H4K16ac+ TEs interact with genes and regulate their expression in cis.

Many epigenetic repression mechanisms have been demonstrated for TEs. However, how TEs are derepressed in a cell type specific manner or how these repressive complexes are counteracted when TEs are upregulated in stem cells or during neuronal lineage specification is yet to be discovered. We show that MSL/H4K16ac axis plays a significant role in counteracting repressor complexes by maintaining activated chromatin domains at many L1 and LTRs.

Data and methodologies: mostly appropriate, but would urge the authors to improve on the quantitative depiction of their genomic/epigenomic data (missing scale bars for heatmaps, y-axis scales, etc pervasively throughout the manuscript).

Thanks for this suggestion; we have now improved the data presentation by including scale bars, axis scales, and peak number details.

Conclusions: the most important test of their observations is Fig 4D, in which they target TE and test if chromatin repression using the Crispr-I strategy results in decreased putative target gene expression. This experiment shows that 6/7 of the genes have lower expression levels after dCas9-KRAB targeting of the TEs—and the effect size is comparable to what we normally expect with this technologies' dynamic range. However, since the gRNA likely targets hundreds or thousands of sites across the genome since L1 and ERV are typically high copy # and similar in DNA sequence, it would be preferable to genetically delete several of these putative enhancers using a genetically clean KO strategy.

We have now used more controls for CRISPRi and successfully performed genetic deletions for candidate L1s in hESCs.

We have also performed genetic deletion of four candidate L1s (heterozygous deletions), which led to a significant reduction in the expression of putative target genes that were predicted based on loops or TADs in which they are located. Notably, other nearby genes we tested were unaffected, demonstrating gene regulation specificity.

References in general are appropriately cited.

Clarity of text: in general, I was able to follow the authors reasoning of experimental logic and interpretation.

Specific comments to the authors

Fig 1 B: ambiguous what the y-axis means for this stacked barplot. Is the height of the bar indicative of the enrichment over the background? Or is the top of the coloured segment the enrichment level?

For clarity, we have now replotted as a dot plot.

Fig 1E: no scale bar on the heatmap. Is this log₂ FC or FPKM?

Heatmaps are CPM (now clarified also in the legends).

Fig 1F: No y-axis scale on the genome browser snapshot. Is this FPKM or log₂ FC? Are all of the browser tracks all on the same scale?

We have improved the data presentation by including scale bars, axis scales, and peak number details. Scale for browser tracks has been added to all figures.

Fig 2C: Western blot should indicate what HERV protein is being detected, ie HERVK, HERVH, HERV-W etc and gag, env, capsid etc. Also would be more appropriate if the western blot showed a histone loading control, preferably pan-H3—since it looks like knock-down of MSL3 leads to an increase in H3K27ac.

We have indicated that the HERV protein name (ERVW1) antibody is raised against. We have not found an increase in H3K27ac upon MSL depletion (See fig S8A). Increased H3K27ac could be due to a slightly higher amount of protein in the MSL knockdown lanes (see western blot for Tubulin in Fig 5C).

The authors note in Fig 2E that the effect of MSL3 on gene expression was minimal, but the figure panel is labeled with “****” indicating that this effect is highly significant statistically. To clarify this statement, maybe the authors would like to amend that statement to indicate that “the fold/change or effect size on gene expression was minimal”.

Thanks for this suggestion; although the effect on gene expression was minimal, we have included a new analysis to show specific effects on genes closer to H4K16ac+ L1 and LTRs compared to genes that are away from genes (new Fig. 5H and I). Please also see the correlation of H4K16ac+ L1 with gene expression level in fibroblasts but not H4K16ac- (new fig S9).

Fig 2D: the effect of MSL3 depletion on ATAC-seq levels should be quantified with a statistical test, not just depicted as heatmap. Typically, we would depict a comparison such as control vs knockdown as a log₂Fold-change of the experimental/control—it is unclear based on the labeling of the figure legend and the panel if this is a subtraction (control minus experimental) operation

and why the authors would chose to display these data in such a non-standard way. There is also no heatmap scale bar on this figure panel—so it is difficult to interpret the absolute vs relative levels of these data. The authors might want to plot this panel using substantively different colors from dark purple and dark blue to help guide the reader that they are viewing Cut-N-tag data vs ATAC-seq respectively.

Thanks for all these suggestions; we have now improved the visualisation by incorporating all the suggestions and changed the ATACseq data to Log2FC (new Fig 5E).

Fig 2 in general: typical to indicate the n=sample size for each genome feature or category, this can be listed in the figure legend or directly on the figure panel itself, but is not NSMB formatting practice to make the reader refer to a supplemental statistics table to obtain this information.

In this version, sample sizes and stats are mentioned for all the figures, and details of the stats are in the figures and legends.

Fig 3: no scale bar on the heatmaps.

Fig 3E: no y-axis scale for these data

Fig 3E: LTR subtypes should be indicated. For instance in 3E, is this an LTR5, LTR5A, LTR5B, LTR5HS?

We now have added scale bars and a Y-axis scale for all the data.

We have added subtypes for LTR5 and other LTRs in the New Fig 2.

Fig 4D: The Crispr-I technique (using a dCas9-KRAB) fusion is an alternative to genetically deleting the putative enhancer (by using unique sequence around the repeat element to specifically delete the L1 or ERV)—and certainly worth trying as a first pass to screen for putative TE enhancer sequences that indeed regulate the cognate gene. However, the effect size is rather small for several of the analyzed genes. How many total putative TE-enhancer: gene pairs were analyzed? Does Fig 4D depict all of the candidates tested in this assay or is this a subset?

We agree with the limitations of CRISPRi and have included more controls (L1s with no acetylations located closer to a gene (USP38) than H4K16ac+ L1s). Our new data shows a specific reduction in the expression of USP38 upon CRISPRi for H4K16ac+ L1s compared to H4K16ac- L1s.

Indeed, the new Figure 4 depicts all the TEs analysed. We found that two of the tested ERV/LTRs closer to TAD borders did not affect the expression of nearby genes (Fig 3D),

suggesting that some of these TEs may not activate the expression of nearby genes. We have now tested a minimum of two genes per TE.

I would urge the authors to pick several of these candidate regions and validate them using crispr/cas9 KO to delete the TE genetically. I understand that with primed hESC, making clonal cell lines can be time consuming and technically difficult, but since the authors have chosen to use a Crispr-I strategy, they are globally targeting many individual instances of L1 or ERVs and it is possible that the effect they see on gene expression of putative cognate genes is either indirect or is blunted by having to recruit the dCas9-KRAB fusion to thousands of sites across the genome, thus giving a rather weak effect size.

We have now performed genetic deletions for 4 L1s (three heterozygous clones and two pool of cells with nearly 50% efficiency in deletion). Our new data from CRISPR deletions is consistent with the findings from our CRISPRi approach, with the exception of the L1 deletion at MOXD1, where CRISPRi led to the downregulation of both MOXD1 and STX7, both of which engage in chromatin loops with the tested L1 region. Interestingly, L1 deletion led to the downregulation of MOXD1 but not STX7. The other three deleted H4K16ac+ L1s are intergenic, and these deletions affected the expression of the same genes that showed downregulation in CRISPRi, but not of other control genes (Fig 4).

The evidence that ERVs or L1 can act as enhancers based on the data they present here is rather weak. For instance, chromatin-looping or chromatin-conformation data (such as MicroC) could help boost evidence to support the claim that L1 can act as enhancers of the MoxD1 gene. In the current form, Fig 4C is essentially un-interpretable as there is no quantification of the interaction strength between the L1 and the Moxd1 gene. Is there a genome-wide analysis of these data that can globally support their claims re: TE-enhancers and genes?

We have now performed genome-wide analysis for loops using micro-c data instead of Hi-C to show the looping between L1 and LTRs with genes. Indeed, the L1 region shows a significant loop ($p < 0.05$) with MOXD1(5' end) and other nearby gene STX7 (Data S1). The genome-wide analysis also revealed that a significant fraction of acetylated TEs engages in a chromatin loop with genes, compared to TEs that lack acetylations (Fig. 3F).

Since intronic L1 in the host gene can have a strong effect on the expression of the CYP3a5 gene (see Liu, et al. Nature 2018 which the authors cite) by recruiting the HUSH repressive complex that also decreased the host gene expression or conversely is the case of this manuscript allowing for MSL recruitment boost expression of genes near H3K122ac/H4K16ac marked L1/ERVs, the

effect of TE on neighboring genes can be difficult to dissect—and are not necessarily a result of TE-comprised enhancers.

We agree that it is challenging to dissect the cis-regulatory effect when the L1 is within the intron of a gene. Hence, we have included CRISPRi and deletions for L1s that are not in the introns but are located within the same TAD, or contact the genes via chromatin loops (Fig 4 G-K). All of these cases demonstrate their cis-regulatory effect. In the case of L1s located within introns of a large gene (MOXD1), we show significant loops from this L1 region with the 5' end of MOXD1 (~70kb from L1) and another nearby gene STX7 (~150kb from L1). It is also possible that H4K16ac at some TEs located at the introns could also facilitate transcriptional elongation of the host genes.

Decision Letter, first revision:

Message: Our ref: NSMB-A46462A-Z

22nd Mar 2023

Dear Dr. Pradeepa,

Thank you for submitting your revised manuscript "H4K16ac activates the transcription of transposable elements and contributes to their cis regulatory function" (NSMB-A46462A-Z). I apologise for the delay in sending you the final decision, it took longer than expected to obtain the referee reports.

The study has now been seen by the original referees and their comments are below. The reviewers find that the paper has improved in revision, and therefore we'll be happy in principle to publish it in Nature Structural & Molecular Biology, pending minor revisions to satisfy the referees' final requests and to comply with our editorial and formatting guidelines.

To facilitate our work at this stage, it is important that we have a copy of the main text as a word file. If you could please send along a word version of this file as soon as possible, we would greatly appreciate it; please make sure to copy the NSMB account (cc'ed above).

Thank you again for your interest in Nature Structural & Molecular Biology Please do not

hesitate to contact me if you have any questions.

Sincerely,

Carolina

Carolina Perdigoto, PhD
Chief Editor
Nature Structural & Molecular Biology
orcid.org/0000-0002-5783-7106

Reviewer #1 (Remarks to the Author):

The authors have addressed most of my concerns and I have no further comment on the manuscript.

Reviewer #2 (Remarks to the Author):

I appreciate the authors' efforts in responding to the previous reviews, and the revisions have improved the manuscript. The authors have addressed most of my technical concerns.

My biggest concern is the biological insights of this paper to the area, I respectfully disagree with the authors' claim for the novelty of their manuscript. For epigenetic regulation, it is known that both L1 5'UTR and ERV contain many histone modifications including both acetylation and methylation (PMID: 30604769) and act as enhancers (PMID: 22905872, PMID: 33602314, PMID: 25319995). And it is also known that L1 engaged in 3D genome chromatin loops (PMID: 32286261). Besides, most studies surveyed the epigenetic regulation mechanism and function of TEs between development or diseased processes (PMID: 32895553, PMID: 36418324), while this manuscript only focused on hESC which lacks deep biological meaning that also limited my enthusiasm.

Reviewer #3 (Remarks to the Author):

I thank the authors for addressing my previous comments. I'm in general satisfied with the revision and new experiments. I suggest the authors tune down many of the claims since most of the evidences are correlative rather than casual. For example, statement such as "These analyses show that TEs enriched with histone acetylations contribute to 3D chromatin folding and looping interactions with genes" are not supported by any data. The data is correlative and does not suggest directional contribution.

Author Rebuttal, first revision:

We have now addressed two remaining issues raised by reviewers 2 and 3. Please see below the point-by-point response to their comments. We have also edited the manuscript as suggested by the editorial team.

Response to comments

Reviewer #2:

I appreciate the authors' efforts in responding to the previous reviews, and the revisions have improved the manuscript. The authors have addressed most of my technical concerns.

My biggest concern is the biological insights of this paper to the area, I respectfully disagree with the authors' claim for the novelty of their manuscript. For epigenetic regulation, it is known that both L1 5'UTR and ERV contain many histone modifications including both acetylation and methylation (PMID: 30604769) and act as enhancers (PMID: 22905872, PMID: 33602314, PMID: 25319995). And it is also known that L1 engaged in 3D genome chromatin loops (PMID: 32286261). Besides, most studies surveyed the epigenetic regulation mechanism and function of TEs between development or diseased processes (PMID: 32895553, PMID: 36418324), while this manuscript only focused on hESC which lacks deep biological meaning that also limited my enthusiasm.

Answer:

We have now amended the text in the introduction to reflect on the previously known histone modifications (line 55).

The discussion covers details of possible neurodevelopmental disease links (MSL3 syndrome).

Reviewer #3:

I thank the authors for addressing my previous comments. I'm in general satisfied with the revision and new experiments. I suggest the authors tune down many of the claims since most of the evidences are correlative rather than casual. For example, statement such as "These analyses show that TEs enriched with histone acetylations contribute to 3D chromatin folding and looping interactions with genes" are not supported by any data. The data is correlative and does not suggest directional contribution.

Answer: We have now changed the text as requested by this reviewer (Line number 184).

Final Decision Letter:

Message 5th May 2023

:

Dear Dr. Pradeepa,

We are now happy to accept your revised paper "H4K16ac activates the transcription of transposable elements and contributes to their cis regulatory function" for publication as a Article in Nature Structural & Molecular Biology.

As soon as your article is published, you can generate your shareable link by entering the DOI of your article here: http://authors.springernature.com/share. Corresponding authors will also receive an automated email with the shareable link

Your paper will be published online soon after we receive proof corrections and will appear in print in the next available issue. You can find out your date of online publication by contacting the production team shortly after sending your proof corrections. Content is published online weekly on Mondays and Thursdays, and the embargo is set at 16:00 London time (GMT)/11:00 am US Eastern time (EST) on the day of publication. Now is the time to inform your Public Relations or Press Office about your paper, as they might be interested in promoting its publication. This will allow them time to prepare an accurate and satisfactory press release. Include your manuscript tracking number (NSMB-A46462B)

and our journal name, which they will need when they contact our press office.

About one week before your paper is published online, we shall be distributing a press release to news organizations worldwide, which may very well include details of your work. We are happy for your institution or funding agency to prepare its own press release, but it must mention the embargo date and Nature Structural & Molecular Biology. If you or your Press Office have any enquiries in the meantime, please contact press@nature.com.

Please note that *Nature Structural & Molecular Biology* is a Transformative Journal (TJ). Authors may publish their research with us through the traditional subscription access route or make their paper immediately open access through payment of an article-processing charge (APC). Authors will not be required to make a final decision about access to their article until it has been accepted. <https://www.springernature.com/gp/open-research/transformative-journals> Find out more about Transformative Journals

Sincerely,

Carolina Perdigoto, PhD
Chief Editor
Nature Structural & Molecular Biology
orcid.org/0000-0002-5783-7106
